# MESSAGEix-GLOBIOM Nexus Module: Integrating water sector and climate impacts

Muhammad Awais[1,2], Adriano Vinca[1], Edward Byers[1], Stefan Frank[3], Oliver Fricko[1], Esther Boere[3], Peter Burek[3], Miguel Poblete Cazenave[1], Paul Natsuo Kishimoto[1], Alessio Mastrucci[1], Yusuke Satoh[3,5], Amanda Palazzo[3], Madeleine McPherson[2], Keywan Riahi[1,2,4], Volker Krey[1]

[1] Energy, Climate & Environment Program, International Institute of Applied Systems Analysis, Laxenburg, Austria
[2] Institute for Integrated Energy Systems, University of Victoria, Canada
[3] Biodiversity and Natural Resources Program, International Institute of Applied Systems Analysis, Laxenburg, Austria
[4] Technical University Graz, Austria
[5] Moon Soul Graduate School of Future Strategy, Korea Advanced Institute of Science and Technology, Daejeon, Korea

*Correspondence to*: Muhammad Awais (awais@iiasa.ac.at)

**Abstract.** The Integrated Assessment Model (IAM) MESSAGEix-GLOBIOM developed by IIASA is widely used to analyse global change and socioeconomic development scenarios within the energy and land systems across different scales. However, until now, the representation of impacts from climate impacts and water systems within the IAM has been limited. We present a new nexus module for MESSAGEix-GLOBIOM that improves the representation of climate impacts and enables the analysis of interactions between population, economic growth, energy, land, and water resources in a dynamic system. The module uses a spatially resolved representation of water systems to retain hydrological information without compromising computational feasibility. It maps simplified water availability and key infrastructure assumptions with the energy and land systems. The results of this study inform on the transformation pathways required under climate change impacts and mitigation scenarios. The pathways include multi-sectoral indicators highlighting the importance of water as a constraint in energy and land-use decisions and the implications of global responses to limited water availability from different sources, suggesting possible shifts in the energy and land sectors.

## 1 Introduction

Multiple inter-sectoral objectives, including economic, environmental, and social goals, are integrated into formulating effective, sustainable policies over the long term. Nexus approaches have been increasingly used and considered in policy analysis, including the Sustainable Development Goals (SDGs), to exploit synergies and avoid negative trade-offs and unintended consequences considering the increased awareness of the interdependencies between the energy-water-land (EWL) sectors. Climate policy assessment helps identify pathways that can help achieve the 'well below 2°C' global warming target and other SDGs, such as access to clean

energy, water and sanitation, and food security (Parkinson et al., 2018; Khan et al., 2017, 2018; Parkinson et al., 2019b). In addition to climate change risks, limited resources compounded by population and GDP growth pose an additional challenge (Byers et al., 2018). Integrated Assessment Models (IAMs) help researchers and policymakers understand the long-term consequences of varying socioeconomic development and climate change scenarios. These scenarios assess the costs and benefits of climate change impacts and mitigation strategies. These models integrate sectors (global economy, energy, water, agriculture, and forestry) to provide policy insights relevant to climate change scenarios (Weyant, 2017). IAMs provide long-term transformation pathways to answer critical questions on climate change transition to ambitious climate policy goals (Riahi et al., 2017).

Substantial efforts have been made to develop scenarios that inform a range of futures with varying societal and socioeconomic assumptions. (Riahi et al., 2017) The most used set of scenarios in IAMs includes the Shared-Socio-economic Pathways (SSPs), a group of five quantified narratives for the evolution of socioeconomic development globally for the 21st century (O'Neill et al., 2017), and Representation Concentration Pathways (RCPs), a set of four scenarios spanning a range of radiative forcing values (van Vuuren et al., 2011). These narratives have been translated into assumptions for economic growth, population change, and urbanization to analyse baseline and climate mitigation scenarios (Riahi et al., 2017).

Although SSPs were designed to analyze the challenges for mitigation and climate adaptation, integration of climate impacts and adaptation of energy and land sectors to water sector constraints has, until recently, been limited in the IAM scenarios due to substantial challenges in technical implementation and representation of climate impacts. Long-term assessment of climate mitigation scenarios often neglects the climate impacts on system performance, resulting in avoiding adaptation costs in the analysis (Calvin et al., 2013; Piontek et al., 2021). IAMs typically operate at a regional or continental scale to inform future pathways, whereas adaptation strategies require a more nuanced, localized focus emphasizing national and sub-national levels (Andrijevic et al., 2023). More detailed information on the spatial distribution of costs and benefits of impacts and adaptation is required to inform adaptation actions and policies (Patt et al., 2010).

Impact modelling activities across diverse modelling groups, such as the Intercomparison Model project (ISIMIP) (Frieler et al., 2017), have been carried out to understand the impacts of climate change better individually. These sectoral assessments evaluate biophysical impacts such as changing yields, runoff changes, food production, and groundwater. Economic impacts are subsequently estimated using various methodologies, chosen based on the specific type of impact considered, such as the correlation between climate damages and temperature variations. Some studies have empirically linked climate conditions with socioeconomic systems and incorporated distributional factors into cost-benefit models, resulting in increased social costs of carbon and more stringent mitigation pathways (Parry and Carter, 2019; Howard and Sterner, 2017). Incorporating the representation of biophysical climate impacts into integrated assessment models is crucial to understand how various sectors influence techno-economic scenarios and to

identify appropriate mitigation and adaptation strategies (van Maanen et al., 2023; Andrijevic et al., 2023). (Piontek et al., 2021) analyzed the economic impacts of climate change using the REMIND IAM model, but biophysical climate impacts were not represented. (Soergel et al., 2021a) emphasized the significance of considering the consequences of climate impacts and evaluating how integrated scenarios respond to these impacts, especially regarding sustainable

development pathways.(Schultes et al., 2021) highlights the economic impact of climate change, advocating for immediate mitigation to reduce long-term damages and align with cost-effective Paris Agreement targets. This study proposes a framework incorporating high-resolution model outputs of biophysical climate impacts into IAMs, strengthening the water sector's resilience, and crafts scenarios with sustainable development objectives to evaluate climate change effects across

various pathways, including mitigation, adaptation, and sustainability.

New analytical approaches and solutions are required to address the challenges of impact and adaptation in long-term policy analysis (Wang et al., 2016; Patt et al., 2010; Riahi et al., 2017). There is a need for a balanced synthesis of Shared Socio-economic Pathways (SSP) narratives with climate impacts, adaptation, and resilience pathways to assess water, food, and energy

security to access sectoral adaptation costs and impacts (Rasul, 2016; Schleussner et al., 2021). Regions highly exposed to climate impacts, highly vulnerable populations (Byers et al., 2018), and developing regions face the biggest challenge in adapting to climate change impacts and simultaneously meeting growing population-driven demands in the EWL sectors (Rasul and Sharma, 2016). Integrating cross-sectoral EWL nexus analysis in IAMs can help identify trade-

offs and synergies, integrate policy implementations, and address equity dimensions, such as the population exposed to hunger or lacking access to sanitation and electricity. This holistic approach is designed to elicit a model endogenous response to climate impacts and SDGs constraints, thereby enhancing systemic resilience and advancing sustainable development. However, it does not delineate specific adaptation policies at the community level. Due to the spatial and temporal

complexity of hydrological data, it is challenging to translate hydrological information into the IAMs. Usually, the spatial extent of IAMs is macro-regions, and the aggregated hydrological information loses adequate information at a macro-level. There is 105 always a need to find a middle ground between showing the hydrological process more accurately and lowering the cost of computing (Fricko et al., 2016; Parkinson et al., 2019b). There have been efforts to link a higher

spatial resolution water sector to account for hydrological balance and constraints in IAMs, such as (Yates, 1997) and (Kim et al., 2016). Addressing the identified gaps, this study proposes a framework that integrates climate impacts with an emphasis on the water sector's role in climate change and develops scenarios in sync with sustainable development assumptions to evaluate the effects of climate change within the contexts of mitigation, adaptation, and sustainable

development pathways.

This paper introduces a new module of the global MESSAGEix-GLOBIOM framework (Riahi et al., 2021; Krey et al., 2016). The Nexus module attempts to fill the gap in integrated assessments by improving the representation of biophysical climate impacts across the EWL sectors and enhancing the water sector representation. We develop scenarios that can effectively

capture climate impacts across multiple sectors using this module. Then, these scenarios are combined with SDG targets in EWL sectors to capture the synergies and trade-offs of climate impacts and sustainable development pathways.

      The manuscript is structured as follows: Section 2 comprehensively explains the module's structure, improvements, and modular procedures, with detail on specific components

of the module, such as the water sector, biophysical climate impacts, Sustainable Development Goals, and flexibility at different scales (with Zambia as an example), described in section 3. Section 4 presents the results of the module's ability to answer different research questions, and Section 5 concludes with a summary of the study's significant findings and contributions.

## 2. Model structure & workflows

Least-cost optimization using engineering-economic modelling is a common approach for long-term energy, water, and land planning (Barbier, 2012; Khan et al., 2017). However, it is not typically performed in a holistic manner that jointly considers system solutions across sectors in a single algorithm. These approaches have been a vital component of the MESSAGEix framework in analysing sustainable transition in climate change mitigation and sustainable

socioeconomic development (Khan et al., 2018; Huppmann et al., 2019). Engineering-economic modelling methods to quantify impacts, resource potential, and costs across different spatial and temporal scales are employed within the nexus module. The approach is both engineering and economical in scope because it combines physically based models of infrastructure systems with cost functions and decision rules for operation, expansion, and retirement at the process level

through time. The theoretical underpinning of decision modelling is that system design choices are made at least cost over the planning horizon in a perfectly foresight, integrated way. The end-user prices for consumers are minimized, and flexibilities across sectors to absorb sectoral trade-offs are fully utilized and planned for in advance.

      The "nexus" module of the MESSAGEix-GLOBIOM framework, MESSAGEix-

GLOBIOM Nexus v1 presented in this paper, contains endogenous spatially- and temporally explicit climate impact constraints and water allocation algorithms. This module extends the foundational work carried out by (Parkinson et al., 2019b). It addresses the gaps in the previous study by improving the water sector resolution, water constraints, and climate impacts. The module here refers to expanding the core global framework of MESSAGEix-GLOBIOM to

represent specific dimensions straightforwardly at the cost of increased computational complexity and cost. The MESSAGEix-GLOBIOM Integrated Assessment framework is a global energy-economic-agricultural-land use model that evaluates the interconnected global energy systems, agriculture, land use, climate, and the economy. The MESSAGEix framework optimizes the total discounted system costs across all energy, land-use, and water sector representations using Linear

Programming. It provides options for both perfect foresight and recursive-dynamic modes. Its adaptability and flexibility make it a powerful instrument for optimizing transformation pathways at various scales, emphasizing minimizing system costs. It comprises five complementary modules: the energy model MESSAGEix (Huppmann et al., 2019), the land use model

GLOBIOM (Havlík et al., 2014), the air pollution and greenhouse gas (GHG) model GAINS, the aggregated macro-economic model MACRO, and the simple climate model MAGICC (Meinshausen et al., 2011). The framework combines the MESSAGEix and GLOBIOM models to assess and model policy scenarios' economic, social, and environmental implications. The framework comprehensively examines the trade-offs and synergies between numerous policy objectives, such as reducing greenhouse gas emissions, boosting food security, and safeguarding natural resources. To access sustainable development targets, the framework is utilized to evaluate the feasibility and implications of alternative policy choices and to guide decision-making.

The nexus module simultaneously determines energy portfolio, land use, associated water requirements, and feedback from constrained resources, such as limited water availability for energy and land use resource usage. It includes a framework for connecting information from hydrological models. It is designed to adapt any Global Hydrological Model (GHM) output and be flexible across different spatial scales (regional definitions, global and country scales). A higher-resolution spatial layer at the basin scale is embedded within the module to retain valuable hydrological data. The information from the water sector is then mapped to the global MESSAGEix energy system at the MESSAGEix native region level. This connects valuable water resource data to the energy sectors and vice versa. The framework balances basin-level water availability and demand while mapping water necessary for energy and land usage at the MESSAGE native region level. The nexus module tracks annual municipal and industrial water demand, water required for power plant cooling technologies, energy extraction, and irrigation water use, balancing through water supply from several sources, such as surface water, groundwater, and desalinated water.

Furthermore, a wastewater treatment infrastructure representation tracks the water during collection, treatment, and reuse. Water demands are tracked across urban and rural components to enable a more comprehensive understanding of future development and adaptation needs. Additionally, biophysical climate impacts are integrated across EWL sectors, including water availability, desalination potential, hydropower potential, air-conditioning cooling demand, power plant cooling potential, and land-use variables (bioenergy, irrigation water) to account for the feedback associated with climate change within the module. GLOBIOM was also adjusted to capture water supply, availability, scarcity, and demand from other sectors based on GHM's hydrological data under different climate-forcing scenarios. In this case, GLOBIOM and the MESSAGEix nexus module are configured to use outputs from gridded GHMs from the Inter-Sectoral Impact Model Intercomparison Project (ISIMIP) (Frieler et al., 2017). This information is specified for 210 river basins based on the Hydro SHEDS basin delineation (Lehner et al., 2006) (Figure 3).

One of the critical features of the Nexus module is its ability to simulate global interactions across multiple sectors and systems. It allows the module to represent the complex feedback and spillover effects from policy interventions, such as the potential implications of land use changes on the global food system and the energy sector or the water footprints of the energy system. The framework facilitates a comprehensive assessment of policy options by integrating

scenario-based projections, including population and economic growth, technological advancements, and resource limitations.

The integrated approach thoroughly considers the trade-offs and synergies between diverse policy objectives, such as reducing greenhouse gas emissions, enhancing food security, and protecting natural resources. Considering biophysical climate impacts across different sectors helps to access different adaptation needs and responses in different sectoral outputs across different pathways. In the context of sustainable development, it can analyse the viability and implications of various policy alternatives and inform decision-making.

The MESSAGEix-GLOBIOM framework allows flexible integration with different modules, such as those on water, transport, materials, and buildings. The development process of the nexus module is divided into four phases: (i) identifying databases and literature studies for key assumptions; (ii) data processing to make the data model compatible; (iii) setting the core module, which compiles the data and populates it into the core model; and (iv) post-processing of the model outputs to provide ready-to-use results in a database and for visualization tools such as IIASA scenario explorer (Huppmann et al., 2018).

The module uses SSP-RCP (Shared Socioeconomic Pathways – Representative Concentration Pathway) combinations as narratives for creating a baseline scenario. Each scenario is developed using SSP-RCP combinations, national policies, and Sustainable Development Goal (SDG) assumptions aggregated at the R11 region, a spatial delineation of 11 global regions used in the MESSAGEix-GLOBIOM. National policies, including energy use and emission trajectories, are formulated based on the existing MESSAGEix-GLOBIOM at 0.5° x 0.5° spatial resolution, distributed monthly over the growing season based on local cropping calendars for a 10-year time step. These requirements are used as input to the GLOBIOM model. We used the Global Hydrological Models (GHMs) outputs from the ISIMIP database (Frieler et al., 2017) for water availability and hydropower potentials for biophysical impact indicators. The GLOBIOM model upscales these water requirements and provides irrigation requirements at an aggregated 37 regions based on land-use allocation decisions.

A typical scenario from the MESSAGEix-GLOBIOM is used to develop and extend the nexus module and consists of several crucial components(Riahi et al., 2021). Socioeconomic assumptions on population and GDP are used to form energy demand projections. Nationally Determined Contributions (NDCs) are applied to various sectors and configurations as policy implications, including but not limited to emission targets, energy shares, capacity or generation targets, and macro-economic targets. The reference energy system in this scenario features a comprehensive set of energy resources and conversion technologies from extraction to transmission and distribution. This scenario's outcome estimates technology-specific multi-sector responses and pathways for various sectoral targets. The analysis is based on the Shared Socioeconomic Pathway (SSP) 2, which builds on historical trends as the starting point. The time horizon for the optimization framework of MESSAGEix-GLOBIOM extends from 2020 to 2100, with a non-regular distribution of time steps.

Further information on the typical scenarios of MESSAGEix-GLOBIOM can be found in (Krey et al., 2016). The scenario is further extended from the typical scenario in the nexus module using certain policy and technological assumptions. The configuration can handle any SSP-RCP combinations to allow access to a diverse range of pathways compared to each other and the Reference scenario.

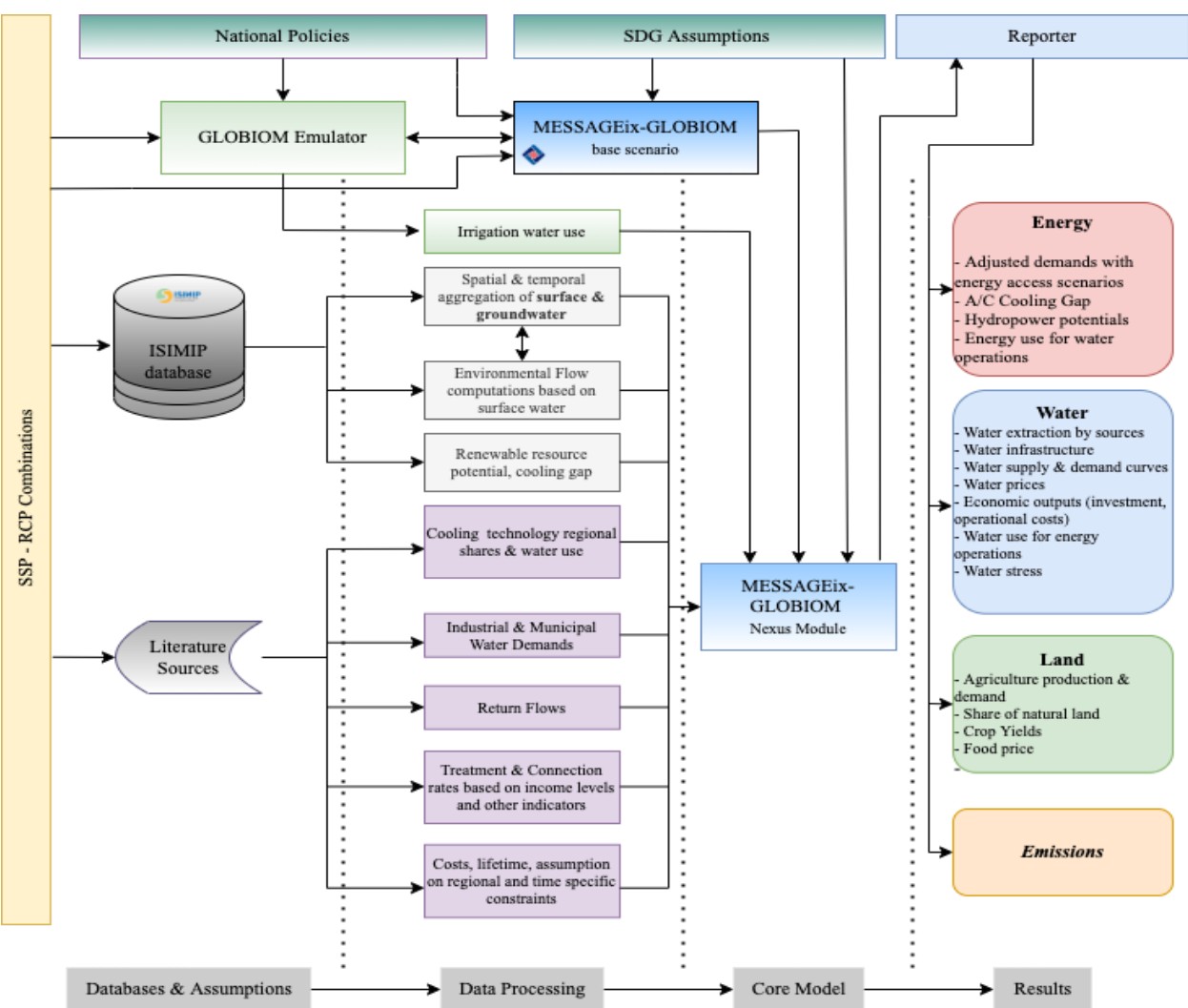

**Figure 1** Structure & data flows of MESSAGEix-GLOBIOM Nexus Module. SSP-RCP combinations of scenarios are used as basis for development of nexus module. The module is built on the typical MESSAGEix-GLOBIOM scenario. The typical scenario has updated biophysical climate impacts in the energy and land sectors and then the water system is added. The database assumptions, structure and processing are the main components of this study besides the core model. Using the computational tools and post-processing methods, multi-dimensional sectoral results inform the pathways for different scenarios.

### 3 Water, Climate, and SDG implementation and results

The subsequent sections explain the modelling framework's water resource structure (supply, demand, and infrastructure) (Section 3.1), and Sections 3.2 and 3.3 discuss integrating biophysical climate impacts and SDG-related assumptions within the module.

.

### 3.1 Water resources and the water sector

The reference system for the water sector in the nexus module of MESSAGEix-GLOBIOM is shown in Figure 3. This study applies the MESSAGEix-GLOBIOM (energy system model) in native R11 global macro-regions via its energy and land systems. The data sources used across the water sector are detailed in Table 1. The water sector loses important spatial information if aggregated on a macro scale. As a first step toward balancing water demand and supply, we have selected the HydroShed River Basin Level 3 (Lehner et al., 2006), intersected with the R11 region and annual timestep, as the ideal standard scale. This spatial layer results in 210 basins (B210, see Figure 2), providing a more powerful depiction of the supply-demand system (Figure 2). The energy demand for water uses and water withdrawals for irrigation and thermal power plant cooling are mapped from B210 to R11. This allows for balancing water supply and demand estimates at a suitable scale where the economic decision incorporates information on all processes, including water availability. We acknowledge that aggregating water needs across vast regions may underestimate the cascading effect of binding water limitations at the local level and the local level adaptation components.

Using further high-resolution basin definitions adds additional complexity to the model due to upstream and downstream interdependence. Our initial 255 effort identifies the primary long-term regional and global drivers of gross imbalances in the supply and demand for water resources. Our ongoing research focuses on determining the most appropriate geographical (grid, sub-basin, or basin) and temporal (daily, monthly, or annual) scales for reconciling water demands and supplies in the global IAM for more robust climate extremes and adaptation needs. To better understand the spatial distribution and water balance of 260 regions, we can look at the Nile River basin, which extends across South Africa and the Middle East (R11 native regions). Due to the overlapping of these two R11 regions, we come up with two distinct spatial units: Nile-Middle East and Nile-South Africa. Now for Nile-South Africa, using proxy indicators such as basin area and the proportion of available water in each basin, we calculate the proportion of renewable water resources available from the Nile and the total water 265 availability in the South African region. This 'downscaled' value plays a crucial role in the model, allowing us to reconcile the available water supply options with the region's varying water demands.

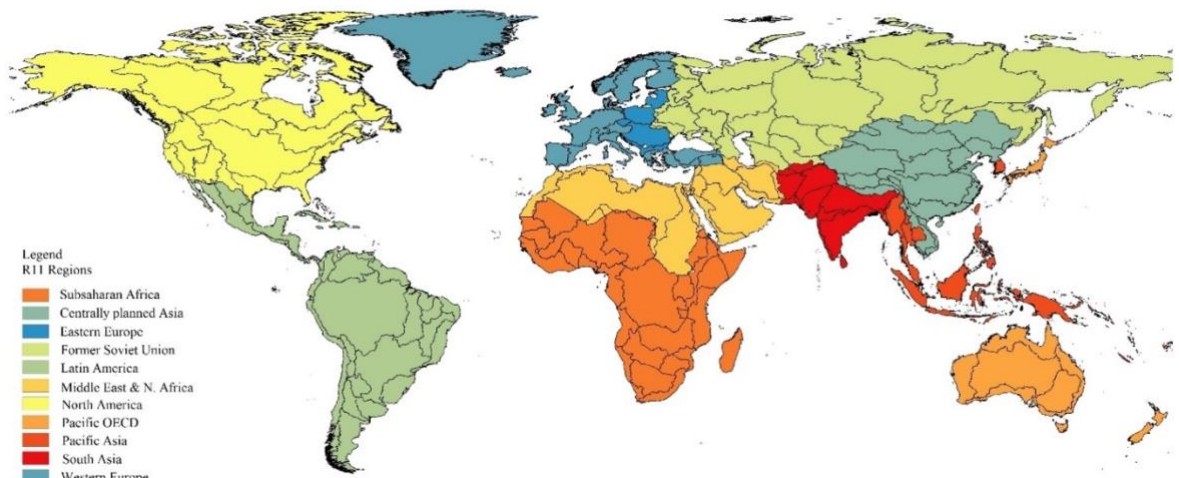

**Figure 2** Delineation of basins within the MESSAGE R11 regions. The HydroSHEDs basin level 3 is intersected with MESSAGE R11 regional delineation, and the new polygon are used as decision units in the water sector. The distinct colors in the maps represent R11 regions however polygons inside each distinct colored R11 regions are the B210 basins intersected by R11 region. The complete list of basin names along with the area in km2 can be looked in the GitHub repository (data/node/B210_R11.yaml)

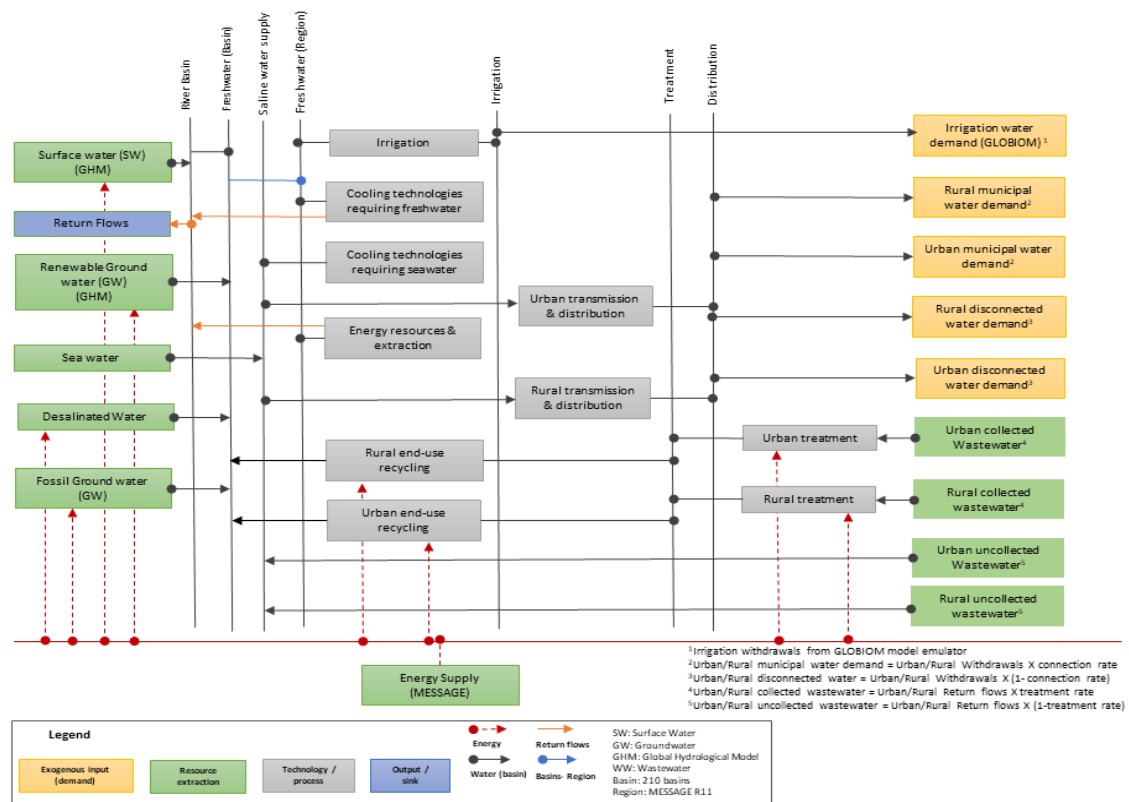

**Figure 3:** Reference System of the water system representation in the nexus module. The arrows show the direction of input/output of different technologies within the framework. Energy footprint of water system is tracked at different supply steps and infrastructure technologies.

The water balance in the water sector of the IAM is

$$Fr_{B,t} + Gw_{B,t} + FGw_{B,t} + Ww_{B,t} + D_{B,t} \geq Mc_{B,t} + \left(Irr_{B,t} + Ew_{B,t}\right) + Ef_{n,t} \qquad (1)$$

$$\left(Irr_{B,t} + Ew_{B,t}\right) \leq \sum\left(Irr_{R,t} + Ew_{R,t}\right) \times share_B \qquad (2)$$

Where *Fr* is the surface freshwater supplied from the river basin, *Gw* is freshwater supplied from groundwater aquifers, *FGw* is the non-renewable groundwater extractions, *Ww* is treated water provided from wastewater recycling facilities, *D* is desalinated water, *Mc* represents municipal and industrial sectoral demands, *Irr* defines the irrigation water withdrawals from the GLOBIOM emulator, *Ew* is the water demand for the energy system. Irrigation and energy water demands are balanced at the regional level, and Ef is Environmental flows calculated using the Variable Monthly Flow (VMF) method (Supplementary Figure S2.1.3) (Pastor et al., 2014).*R* represents MESSAGE energy regions. In contrast, *B* represents river basins within the given MESSAGE regions, and *t* is time periods at a 5-year annual time interval. *share* is the share of freshwater in basins (*B*) per region (*R*) used as a proxy to balance irrigation and energy demands at the basin (B*).* All the values are in km3/yr. In GLOBIOM, irrigation water withdrawals are treated as residual claimants, with the water demands for municipal and energy taking priority (Palazzo et al., 2019; Frank et al., 2021). The water withdrawals are balanced with the supply of each model decision-making period and region.

Within the module, the choice between the supply system is motivated by the associated investments and operational costs. Renewable surface and groundwater freshwater are prioritized based on the cost. The other priority choice of supply between wastewater reuse, desalination, and fossil groundwater varies across regions, and the available potential in each region varies. On the supply side, we use global gridded runoff and groundwater recharge data from the Community Water Model (CWatM) (Burek et al., 2019) and GHM outputs from ISIMIP (Frieler et al., 2017). Three bias-corrected meteorological forcing data from different climate models (GFDL-ESM2M, HadGEM2-ES, IPSL-CM5A-LR) are used to estimate surface runoff and groundwater recharge. We use multi-model ensemble mean runoff and groundwater recharge as an available renewable freshwater resource. We aggregate the gridded data (0.5° X 0.5° spatial, daily timestep) onto the B210 basins and 5-year annual average. For spatial aggregation, the spatial sum is used to sum the grid hydrological outputs (runoff and groundwater recharge) to the B210 basins. The detailed process has been summarized in Supplementary Table S2.

We apply a quantile approach with monthly freshwater (surface and groundwater) resources for temporal aggregation to incorporate hydro-climate variability and prolonged dry periods. For example, for the 10th percentile, the monthly mean is first calculated from daily data. Then, we use the 10th percentile (Q90) of monthly freshwater runoff for a 20-year rolling window to determine a reliable flow for 90% of the time. This type of percentile methodology applied to multi-decadal periods is frequently used in water resource and environmental flow assessments (Prudhomme et al., 2014; Satoh et al., 2022; Gleeson and Richter, 2018) to account for the seasonal low flows experienced in typical wet and average years, although not the driest 10% of months (over 20 years). Figure S2.1.2 shows the Q90 flows overlayed on the monthly flow data

for the significant basins to show their reliable flows. We have run the scenarios for testing the model's sensitivity based on the flow quantiles.

We followed the methodology by (Graham et al., 2020) to estimate the municipal water demands, where urban and rural components are derived from gridded population and income-level projections based on the SSPs, as detailed in(Wang and Sun, 2022). Manufacturing demands are generated following a similar approach used by (Hejazi et al., 2014). Historical country-level data for 2015 is estimated by subtracting energy sector withdrawals from total industrial sector withdrawals. Future changes in manufacturing demands are projected, assuming convergence towards a log-linear model between GDP and manufacturing withdrawals. Demands are distributed across countries based on growth in GDP and then downscaled to 7.5 arcminutes and re-aggregated at the B210 basins. Supplementary Figure S shows urban and rural components of municipal demands and industrial demands for 2050, whereas the data is provided in the GitHub repository (See Data Availability). Supplementary Figures S3.1 & S3.2 show average municipal and industrial demands across the basins. The wastewater treatment system is adapted and improved from the previous implementation by (Parkinson et al., 2019b). Figure 3 shows the framework's conversion steps from wastewater collection to wastewater reuse. The module includes two generalized urban wastewater treatment technologies to simplify the number of decision variables. The first represents a standard secondary-level treatment facility commonly found in a mid-sized city.

In contrast, the second includes recycling capabilities and is parameterized to represent a standard facility suitable for upgrading municipal or manufacturing wastewater to potable standards, such as a membrane bioreactor. In addition, the module includes a rural wastewater treatment technology that meets the United Nations guidelines for clean water and sanitation in rural areas and is equivalent to a standard septic system. It ensures enough wastewater treatment capacity, including recycling and conventional treatment, to support the projected return flow connected to treatment. The desalination potentials have been estimated following the approach in (Parkinson et al., 2019b), where desalination capacity data are inferred against GDP trends using a logistic function. Here, data on water stress from (Byers et al., 2018) have been added to the function to include the climate dimension in the projections (see Figure S 4.1.4).

We use the approach detailed by (Fricko et al., 2016) to calculate water withdrawal and return flows from energy technologies. Each energy technology requiring water is provided with a withdrawal and consumption intensity (e.g., cubic kilometres per GWh). This allows the module to translate technology outputs into water requirements and return flows, which balance with the available supply. For power plant cooling technologies, where the water requirements are calculated as a function of heat rate, the efficiency change in the energy technologies (e.g., lower heat rates) impacts the cooling requirements per unit of electricity produced. The withdrawal and consumption intensities for power plant cooling technologies align with the range reported by (Meldrum et al., 2013a). In contrast, the electricity balance computation includes additional electricity demands from recirculating and dry cooling technologies. Other technologies adhere to the data (Fricko et al., 2016)

The energy footprints of various components of the water sector, including supply (surface water and groundwater extraction), distribution (urban and rural), and wastewater treatment (treatment, recycling, and re-use), are interconnected with the electricity needs of the energy sector. This connection is established through basin-region mapping, which enables the spatial aggregation of appropriate fractions of electricity requirements to the region (R11) where the water sector's electricity consumption is managed. Table 1 indicates different references used for electricity requirements per unit of water infrastructure activity at different stages.

**Table 1:** Data sources used for various parameters and input variables

| Parameter | Description | Data |
|---|---|---|
| **Basin boundaries** | Basin boundaries used from the HydroSheds database (Lehner et al., 2006) to create new spatial units in the water sector | All the processed files are available in the GitHub repository in CSV format(~data/water/delineation) |
| **Power plant water use** | All power plants' water use and investments (Meldrum et al., 2013b) are updated based on the latest powerplant database from Platts (Platts Market Data – Electric Power | S&P Global Commodity Insights, 2022) | All the processed files are available in the GitHub repository in CSV format(~data/water/ppl_cooling_tech) |
| | Hydropower use and investments (Grubert, 2016) | |
| | Parasitic electricity requirements (Dai et al., 2016) | |
| **Water Availability** | Regional shares of cooling (Raptis et al., 2016) | |
| | Runoff & groundwater recharge from the GHM CWatM model (Burek et al., 2019) outputs of the ISIMIP project (Frieler et al., 2017). The outputs are spatially and temporally processed for further use. | |
| | We use groundwater abstraction data from (Wada et al., 2014)and historical water withdrawals from (Wada et al., 2016) to parameterize the historical groundwater extraction. The fraction of groundwater abstraction to the overall withdrawals determined the 'groundwater fraction.' This value is then used on the actual historical water demands included in the model to set the amount of pumping capacity for the future horizon. | All the processed files are available in the GitHub repository in CSV format (~data/water/water_availability) |
| | For the cost of groundwater pumping, depending on the aquifer depth, we use groundwater aquifer depth data (Fan et al., 2013) and energy consumption values from (Vinca et al., 2020) and (Liu et al., 2016). | The energy consumption values vary regionally based on the groundwater table depths. Thus, the processed file is available in the GitHub repository in CSV format (~data/water/water_availability) |
| | Freshwater Energy consumption per unit of water (Liu et al., 2016) | 0.01883 (0.0011 - 0.03653) kwh/km3 |
| | Techno-economic values from (Vinca et al., 2020) and (Burek et al., 2018) | Investment costs are assumed for the whole world. groundwater infrastructure: 155.57 million USD/km3, surface water extraction:54.52 million USD/km3 |
| **Water demands** | Municipal water demands are spatially and temporally processed using the approach followed by (Wada et al., 2016) and using recent and updated data. | All the processed files are available in the GitHub repository in CSV format (~data/water/water_demands) |
| | Irrigation water demands are used from the GLOBIOM model for a set of scenarios aimed at | GLOBIOM Emulator |

| | | |
|---|---|---|
| | achieving multiple, different SDG goals (Frank et al., 2021) | |
| | Treatment & access rates are re-calculated using the approach described in (Parkinson et al., 2019b) and using additional dependent variables in the regression analysis. These treatment and access rates are then used with the return flows from (Wada et al., 2016). | All the processed files are available in the GitHub repository in CSV format (~data/water/water_demands) |
| **Water Infrastructure** | Water distribution & wastewater treatment energy footprints are used by (Liu et al., 2016) | |
| | An upper constraint on desalination potential is implied in the model using multiple regression parameters (GDP, Water Stress Index (Byers et al., 2018), Governance (Andrijevic et al., 2020), and distance to the coast. We use the Desal Data dataset (Global Water Intelligence, 2016) to evaluate the existing (or historical) capacity of desalination units worldwide, gathered at the BCU level. | All the processed files are available in the GitHub repository in CSV format (~data/water/water_infrastructure) |

### 3.2 Climate Impacts

The following climatic impacts are covered in the nexus module and this study: Changes in crop yield, variations in precipitation patterns and drought severity, renewable energy potentials, cooling and heating energy demand, desalination potential, and cooling water discharge for energy use. Impacts on biodiversity are partially included in the evaluation whereby natural land serves as a high-level proxy indicator for the level of biodiversity. This method covers land-use change-induced consequences, which are the primary cause of biodiversity loss in the short term, but excludes direct climatic impacts. Thus, it primarily reflects the consequences of climate and SDG policies. All impact data is derived from the Intersectoral Model Intercomparison Project (ISIMIP) (Frieler et al., 2017) to maintain internal consistency across all indicators and models. The remainder of this section describes the model-specific representation of biophysical climate impacts across the energy, water, and land sectors and the methodological steps required to implement or update new climate impacts. We use the data for RCP2.6 and RCP6.0 to consider the climate impacts, i.e., emission pathways reaching 2.6 $W/m^2$ and 6.0 $W/m^2$ forcing levels in 2100. We have not included GDP and labour productivity implications to focus solely on biophysical impacts.

**Table 2** Summary of biophysical impacts

| Biophysical climate impacts | Approach |
|---|---|
| **Renewable supply (hydro)** | Different costs supply curves based on 0.5x0.5 grid calculations (Gernaat et al., 2021) |
| **Heating/cooling demand** | Impact via population-weighted heating and cooling demands based on the work of (Mastrucci et al., 2021; Byers et al., 2018) 0.5 x 0.5 grid |
| **Water availability** | Runoff and groundwater recharge from CWatM calculated at 0.5 x 0.5 grid (Burek et al., 2020) |

| | |
|---|---|
| **Crop yields** | Climate impacts on crop productivity, nitrogen, and irrigation from the CMIP6 projections of the crop-model EPIC-IIASA are used in GLOBIOM. EPIC-IIASA estimates the impact of climate on rice, maize, wheat, and soy, which are accordingly mapped to the crops in GLOBIOM following (Müller and Robertson, 2014) |
| **Cooling technology capacity factor** | Climate impacts on cooling water discharges for cooling technologies of fossil power plants are used (Yalew et al., 2020) |
| **Desalination potential** | Desalination potential climate impacts are based on water stress outputs from the combinations of GHMs & GCMs (Byers et al., 2018) |

The climate impacts on hydropower energy supply have been based on (Gernaat et al., 2021). The difference between current and projected spatially explicit climate parameters is translated into spatially explicit energy supply estimates, translated to regional cost-supply curves. The climate data were used as input to calculate hydropower potential. It includes the theoretical

potential of the upper limit of resource availability based on physical and hydrological conditions. The climate impacts were calculated for the historical and future periods using the ISIMIP database. The maps of technical potential, combined with economic information, have been used to generate cost-supply curves. These curves show the cumulative technical potential against the production cost, showing that each location's production cost depends on its productivity. Cost-

supply curves are widely used in IAMs to model the long-term cost development of renewable energy technologies. These curves indicate resource depletion, as the most productive sites are slowly being depleted, and thus, higher cost-incurring sites need to be used. On the other hand, note that climate impact on non-hydro renewables is not included in this study because excluding non-hydro renewables in the IAM is not expected to lead to significant discrepancies between the

scenario results. (Gernaat et al., 2021) Have presented relatively small impacts on renewable energy supply.

Regional cooling and heating demand days are based on the dataset and study by (Byers et al., 2018)(Byers et al., 2018), who derived their climate data from an ensemble of downscaled and bias-corrected global climate models (ISIMIP2). The data represents gridded global surface

air temperature data at the daily resolution, summarised to decadal timesteps and a monthly mean and subsequently aggregated to countries, weighted by SSP population. In this study, to estimate the corresponding energy demand in socioeconomic, technology, climate, and policy scenarios, we used two modules within the MESSAGEix-Buildings framework: CHILLED (Cooling and Heating gLobaL Energy Demand model), a bottom-up engineering model to estimate residential

space heating and cooling energy demand; and STURM (Stock TURnover Model of global buildings), a stock turnover model based on dynamic material flow analysis (MFA) to assess the future evolution of the building stock (Mastrucci et al., 2021)(Mastrucci et al., 2021). The resulting estimates of the country's energy demand for cooling for SSP2 under RCP2.6 and RCP6.0 and the assumption of fixed historical temperature are aggregated from the country to the

MESSAGE region. They are added to the module as a subcategory of the residential demand (Figure S5).

Climate impacts on agriculture and assessment of future hotspots are assessed in GLOBIOM by systematically integrating crop yield information from EPIC (Balkovič et al., 2014)(Balkovič et al., 2014) (run for the different GCMs) for 4 crops (corn, wheat, maize, and rice) and applying it using some assumption to our other crops (Jägermeyr et al., 2021)(Jägermeyr et al., 2021). IIASA's Global Forest Model (G4M) is used to model forest growth as a response to climate (Kindermann et al., 2008)(Kindermann et al., 2008). The G4M uses a dynamic net primary productivity model to consider how growth rates are affected by changes in temperature, precipitation, radiation, and soil properties. G4M works with a monthly step, and the highest spatial resolution is 1 km2. The model estimates the impact on net primary productivity, mean annual increment, standing biomass, and harvestable biomass. Factor changes of mean annual increment and biomass accumulation under a certain degree of climate change compared to a no climate change scenario are multiplied by the default rates in GLOBIOM GLOBIOM's biophysical model incorporates agricultural yield, input requirements, and water availability for irrigation from the CWatM. This integration allows us to evaluate the relative effects of climate change on production, consumption, and market conditions and the autonomous adaptation to the impacts resulting from the GLOBIOM. Irrigation water withdrawals from the GLOBIOM are then linked to the nexus module, which balances the water system across other uncertainties.

### 3.3 SDGs

This section describes the energy, water, and land SDG measures in the module, which align with SDG2 (Zero hunger/food access), SDG6 (Clean water and sanitation/water access), SDG7 (Affordable and clean energy/energy access), SDG15 (Life on land/biodiversity). SDG13 (Climate action) is also implicitly included in the framework when emissions constraints are included in the scenario design. In this study, SDG13 is represented by achieving a 2.6 W/m2 (or a well below 2 degrees) target in 2100. This is essentially the goal of the SDG, limiting climate change following the Paris Agreement. Table 3 provides an overview of all the (non-climate) nexus SDG measures, their representation in the modules, and the indicators to measure progress. The main criteria for including measures have been: 1) They should maximally benefit the overall goal, 2) They should be unambiguous and quantifiable, and 3) They should allow for consistent implementation across modules. The interaction between these measures and the other SDG categories is relatively limited.

The MESSAGE-Access-E-USE (end-use services of energy) model (Poblete-Cazenave and Pachauri, 2018; Poblete-Cazenave et al., 2021) is used for the analysis of households' energy access to modern energy services for heating and cooking and has already been used on a global level to study demand in different socioeconomic pathways (Poblete-Cazenave and Pachauri, 2021; Pachauri et al., 2021). An estimation model takes as input micro-level data from nationally representative household surveys covering different regions of the world to estimate behavioural preference parameters that explain the choices of appliances and energy demands for different

end uses based on household socioeconomic and demographic characteristics. Then, a simulation module uses the preference parameters estimated in the first module and additional external drivers that present potential pathways of socioeconomic growth and energy prices to simulate future appliance uptake and household energy demand under each scenario. This process is not internalized in MESSAGEix-GLOBIOM, but instead, a first iteration is performed to estimate the share of the population with access to modern energy sources for cooking (as opposed to traditional biomass or kerosene) given a fixed GDP pathway (SSP2) and energy prices related to each policy scenario. The model also assesses the implication of additional SDG policies regarding costs and transformations in the demand for energy. This is, however, separated from the solution of MESSAGE because an iterative procedure would alter the GDP pathways in the macroeconomic component of the model (MACRO).

The SDG6 narrative is incorporated by applying supply and demand-side development across the water system. The supply-side measure includes constraints on available surface water as environmental flows. Maintaining environmental flows in rivers is instrumental in achieving SDG target 6.6, which aims to protect and restore water-related ecosystems, encompassing a range of natural landscapes from mountains and forests to wetlands, rivers, aquifers, and lakes. We use the Variable Monthly Flow (VFM) method (Pastor et al., 2014) to constrain the monthly surface water available for human use based on environmental flow requirements (EFRs) for wet and dry seasons (Pastor et al., 2014). This method implies that w(ater withdrawals cannot exceed the available residual supply after considering the EFRs. Some regions may be unable to adapt environmental flow targets in 2030 based on historical trajectories due to high withdrawals or fewer governance capabilities. We categorized these basins based on the development status of countries specified by the World Bank, implemented a lower environmental flow target in the respective regions from 2030 onwards and increased the target till 2050, thus following the trajectory of basins with high adaptive capacity. These environmental flow targets also vary across climate impact scenarios. It enables assessing the response to mitigating future demand growth.

The demand-side measures for SDG6 in the water system include targets for reaching sustainable water consumption across all sectors. We constrain the capacity of the water infrastructure system for integrating water access and quality targets. The connection and treatment rates are endogenized in the withdrawals and wastewater collection. These rates are changed to allow shifts in water withdrawals for universal piped access. Wastewater treatment capacity is increased to treat half of all the wastewater collection in the infrastructure system. The connection and treatment rates are adjusted for the basins that can readily adapt; the targets for 2030 are assigned to the basins with more adaptive capacity than those with less adaptive capacity. Increasing the fraction of wastewater treatment also helps to protect ecosystems related to water, thus contributing to achieving SDG6 target 6.6. The rates are projected in the baseline (non-SDG) scenario using a logistic model by combining income projections fitting to national historical data using the approach described in (Parkinson et al., 2019b).

The irrigation conservation approach is implemented to reduce the irrigation withdrawals and reallocate water to other sectors, thus contributing to target 6.4 (Frank et al., 2021). (Pastor et al.,

2019) mentions how the reduced water approach in the irrigation sector in the GLOBIOM model accounts for environmental flows, and the water is reallocated to the environment and domestic uses by saving from the irrigation sector. The module chooses the irrigation water withdrawals based on the land-use emissions and associated costs to keep the land-related trade-offs with water and energy intact through the GLOBIOM emulator. The module enhancements do not cover all SDG6 targets, such as flood management and transboundary cooperation across basins. Concerning biodiversity protection, the GLOBIOM model assumes increased efforts and a doubling of the AICHI Biodiversity Target 11 (e.g., increase the total surface of protected areas to 17% by 2030 (Bacon et al., 2019). In addition, we use the UNEP- WCMC Carbon and Biodiversity Report (Kapos Ravilious C. et al., 2008) to identify highly biodiverse areas and prevent their conversion to agriculture or forest management from 2030 onwards. We consider the area highly biodiverse where three or more biodiversity priority schemes overlap (Conservation International's Hotspots, WWF Global 200 terrestrial and freshwater ecoregions, Birdlife International Endemic Bird Areas, WWF/IUCN Centre of Plant Diversity and Amphibian Diversity Areas).

We estimate residential cooling gaps as the extent of the population needing space cooling without access and the additional energy demand required to close this gap and provide essential cooling comfort to all (Mastrucci et al., 2019). Minimum cooling requirements are calculated under the assumption of durable housing construction and conservative per-capita floor space and cooling operation to provide decent living standards (Kikstra et al., 2021), assuming the gap is covered with current cooling technologies, including fans and AC.

**Table 3**: SDG measures and indicators. Where possible and relevant, measures are fully implemented in 2030 and maintained until 2100 (see this link for SDG description)

| SDG | Measure | Indicators |
|---|---|---|
| **SDG 2 FOOD** | - < 1% undernourishment goal by 2030<br>- Decrease animal calorie intake to 430 kcal/capita/day by 2030 from current levels in overconsuming countries (USDA recommendations for healthy diets) | - Food production<br>- Food prices<br>- Population at risk of hunger |
| | - 50% reduction in food waste compared to SSP2 assumptions | - Food production<br>- Food prices<br>- Population at risk of hunger |
| **SDG6 Water** | - Limited irrigation water withdrawals to sustainable removal rates that do not jeopardize ecosystem services and environmental flows (Frank et al., 2021) | - Water withdrawal (irrigation) |
| | - Based on the variable monthly flow (VMF) method developed by (Pastor et al., 2014), 60% and 30% of the mean monthly natural flow are reserved for ecosystems in low and high flow periods, respectively. | - Water and environmental flows |
| | - A minimum of half of all return flows will be treated by 2030 for | - Population with access to clean drinking water |

| | | | |
|---|---|---|---|
| | | developed regions and 2040 for developing regions. | |
| SDG7 Energy | - | Results from the MESSAGEix-GLOBIOM are iterated through the MESSAGE-Access-E-USE (end-use services of energy) model by the provision of access targets based on income levels and GDP pathways and population with access to modern energy access and the energy demand adjustments are calculated. | - Energy prices<br>- Population with access to modern energy services |
| | - | 90 % access target to modern cooking energy for cooking by 2030 | - Energy prices<br>-Population cooking with traditional biomass |
| SDG15: Life on land | - | Based on (Frank et al., 2021), the expansion of protected lands to 34% in 2030 was assumed, and highly biodiverse areas were identified based on the UNEP-WCMC Carbon and Biodiversity Report (Kapos Ravilious C. et al., 2008) their conversion to agriculture or forest management from 2030 onwards was prohibited. | - Natural land area |

### 3.4 Flexibility across scales

As mentioned in section 2, the module is flexible to adapt to a different spatial dimension with a higher resolution. In this case, we tested downscaling the global module for a particular country, Zambia. The energy sector is downscaled using the country model generator, which is used for various country-scale energy sector analyses, e.g., (Orthofer et al., 2019). However, the nexus module also allows the water system to be prototyped rapidly for a country/basin level. The water reference system described in previous sections is pre-processed onto the higher-resolution spatial units from the gridded datasets, and a base scenario is produced. The workflow diagram to produce the country scale module is shown in supplementary Figure S6. The Zambian scale module is being used to develop an integrated platform combining different high-resolution sectoral models (Water Crop Evapotranspiration model to estimate crop water demand for different crops (Tuninetti et al., 2015), an electricity demand assessment platform, M-LED for communities without electricity supply (Falchetta et al., 2021), OnSSET tool to assess least-cost electrification technologies and investment requirements based on electricity demand and energy potentials (Korkovelos et al., 2019). (Falchetta et al., 2022) discusses the application of such linkages and further details.

### 4 Results

In our analysis, we have currently applied the SSP2 framework in conjunction with both RCP2.6 and RCP6.0 to establish the current module setup. Future work will incorporate a broader array of SSPs paired with various RCPs to ensure a more comprehensive and coherent set of assumptions across different scenarios. Our examination of the biophysical effects of climate change on energy, water, and land use sectors involved contrasting scenarios that integrate climate

impacts—specifically designated as Impacts, Impacts-EN (focusing on the energy sector), Impacts-WAT (water sector), and Impacts-LU (land use)—alongside SDGs. We measured these against a Reference scenario, which is predicated on historical climatic patterns and excludes any projections of climate impacts or SDG considerations. The scenario assumptions are detailed in Table 4.

Our study presents detailed results of water balance flows, providing a critical examination of global water management and the interdependencies within the water, energy, and land nexus. By comparing our module's outputs with benchmark values from the literature, we establish a validation baseline for EWL indicators, ensuring our findings resonate with recognized global estimates. Our study allows the monitoring of water balance flows at varying stages, offering an in-depth understanding of global water management and the intricate nexus between water, energy, and land. These interactions are depicted in Figure 5a in form of a Sankey diagram, along with input details and assumptions expounded in Section 3.1. The module provides a nuanced perspective, capturing the complexities of water resources and their utilization at both global and basin scales. Compared to the literature, global water resources (total runoff) are in the range of approximately 47,220 km3/yr., aligning with those reported by (Burek et al., 2020) and (Sutanudjaja et al., 2018). Across our module's scenarios, water withdrawals or water extractions fell within the 3365–3656 km3/yr., echoing figures found in established literature (Table 5). In our module, global wastewater collection is considered an exogenous input, quantified at approximately 310 $km^3$/yr for 2020, a figure that is broadly in line with the estimates from(Jones et al., 2021). Global wastewater treatment volumes range from 156 to 172 $km^3$/yr, in close agreement with the 187 km3/yr reported by(Jones et al., 2021). For agricultural withdrawals, an essential water use sector, Our module's estimate for agricultural water withdrawals is 2670 km3/yr, which surpasses the 1250–2000 km3/yr range reported by (Burek et al., 2020), yet it is quite consistent with the 2735 km3/yr figure suggested by (Sutanudjaja et al., 2018). Figure 5b shows a range of water supply portfolios with varying water demands. Even though renewable energy sources are crucial overall, the makeup of these portfolios shows significant regional variation (supplementary sections S3 and S4). Characterizing supply portfolios across various river basins will be the focus of future research projects under varying scenarios and water supply reliability levels. However, this structure allows us to see the water management portfolios linked with the energy and land sectors under varying climate and sustainable development scenarios.

**Table 4** Summary of Scenario assumption

| *Scenario* | **Climate Scenario** | **SDGs** |
|---|---|---|
| *Reference* | Historical climate assumptions for RCP 6.0 across EWL sectors. | Not included |
| *Reference (Mitigation)* | Historical climate assumptions for RCP 6.0 across EWL sectors. | |

|  |  |  |
|---|---|---|
|  | *This scenario, although is practically not feasible it is used to compare the responses of the new features* |  |
| *Mitigation* | RCP 2.6 (biophysical impacts of EWL sectors as outlined in Table 2 and section 3.2 ) |  |
| *Impacts* | RCP 6.0 (biophysical impacts of EWL sectors as outlined in Table 2 and section 3.2 ) |  |
| *Impacts_LU* | RCP 6.0 (biophysical impacts of land sector, e.g. crop yields) |  |
| *Impacts_WAT* | RCP 6.0 (biophysical impacts of hydrology) |  |
| *Impacts_EN* | RCP 6.0 (biophysical impacts of energy, e.g., cooling demand and renewable potential) |  |
| *SDGs* | RCP 6.0 (biophysical impacts of EWL sectors as outlined in Table 2 and section 3.2 ) | SDG 2, 6, 7, 13, 15 – as outlined in Table 3 and section 3.3 |

Sectoral withdrawals primarily drive water extraction by source, with irrigation withdrawals from the GLOBIOM model making up a sizable portion. Figure S4.1.3 depicts the outlook for water extraction under the reference scenario. The effects of climate on crop yields show variability, with sugar crops experiencing a significant impact at 16%, while cereals exhibit a comparatively modest change of approximately 1%. The net yield effect is directly influenced by the intensity of nitrogen and phosphorus fertilization, which enhances water use efficiency and consequently reduces the demand for irrigation water. Furthermore, in our climate impact scenarios, increased $CO_2$ levels also increase crop yields and contribute to improved water use efficiency, which is factored into our results. However, these results require cautious interpretation because our study did not account for cultivar optimization. The results affect water withdrawals and consequently influence the portfolio of water supplies. It is essential to highlight the role of enhanced irrigation efficiency assumptions in the SDG scenario, which results in a 29% average reduction in total water withdrawals compared to climate impacts concurrent to the study by (Frank et al., 2021). In addition, these effects contribute to a 28% decrease in the marginal price of potable water due to adaptive responses to climate change impacts in electricity and irrigation withdrawals. In contrast, pursuing the SDGs can result in a significant price increase due to increased allocation to environmental flows.

The results demonstrate that renewable surface water and groundwater are limited and vary across different climate scenarios. These effects decrease renewable water consumption, which is more evident in the land than in the water sector. In addition, our module indicates an increase in the use of alternative water sources such as brackish water, effluent, and desalination in certain regions, indicating that renewable water resources are limited in these areas. These observations

highlight the significance of the SDGs further. For instance, when aligned with SDG 6 targets, the module estimates a 24% reduction in water consumption, resulting in a more sustainable water allocation to environmental flows (Figure 4).

Figure 5 presents a comparative analysis of key Energy-Water-Land (EWL) indicators across a spectrum of modeled scenarios. The boxplot distributions visually depict selected model output indicators for the period from 2030 to 2080, covering scenarios such as Reference, Impacts, Impacts_LU (land use), Impacts_EN (energy), Impacts_WAT (water), and SDGs. The graph's constant trend in energy-related metrics across scenarios stands in stark contrast to the pronounced unpredictability of non-renewable water usage, suggesting that energy indicators are less vulnerable compared to water and land.

Figure 5 also shows that, despite the biophysical impacts, agricultural production doesn't vary much. The SDG scenario, however, results in a considerable 20% decrease in agricultural output, with the biophysical implications of land usage having a particular influence on sugar crop yields. This noteworthy effect emphasizes how susceptible some crops are to changes in land use and how crucial it is to take these effects into account when developing agricultural plans and policies. Furthermore, the primary cause of the decrease in water withdrawals is the consequences of land use, wherein $CO_2$ fertilization effects are a major factor. These effects on land usage decrease the overall need for irrigation and increase the efficiency with which agricultural operations use water. Additionally, the figure also indicates that the cost of potable water has increased by 80%, primarily due to the adoption of environmental flow allocations aimed at protecting freshwater ecosystems and the increased expenses linked to sophisticated wastewater treatment procedures. These elements highlight the intricate relationship that exists between water resource management and economic results as well as environmental care. The geophysical features and land use influences of various regions mostly determine the global consequences of climate change on the water sector, with certain areas experiencing gains while others may have negative effects. Adaptive responses to climatic impacts reduce the number of people exposed to hunger by an average of 11% according to the study. This is not as significant as the 30% reduction in the SDG scenario, which is based on specific actions to reduce the risk of hunger.

It is imperative to exercise caution when interpreting the outcomes of the different scenarios, considering their reliance on several assumptions and their suitability for particular geographical and temporal circumstances. However, these results offer insightful information about the possible financial effects of various water management techniques. Different modeling methodologies may produce different results because assumptions, data inputs, and other elements are inherently variable. It is feasible to determine the most effective and successful tactics and to obtain a more thorough understanding of the probable consequences of different water management systems by comparing the outcomes from many models.

**Table 5** Comparison of EWL indicator results for the year 2020 with published literature sources for module validation.

| Variable/Indicator | Module Value 2020 | Comparison with other studies |
|---|---|---|
| *Primary Energy (EJ)* | 595-599 | 613 (GCAM5.3_NAVIGATE); 591 (IMAGE 3.2); 570 (REMIND-MAgPIE 2.1-4.2) ;575 (MESSAGEix-GLOBIOM_1.1) (Harmsen et al., 2021) |
| *Energy Supply Investments* | 1325-1401 | 1148.13 (IMAGE3.2); 1036/41 (MESSAGEix-GLOBIOM_1.1); 1208 (REMIND-MAgPIE 2.1-4.2) (Harmsen et al., 2021) |
| *Agricultural Production* | 3350 | 4400.6 (IMAGE3.2); 4045 (MESSAGEix-GLOBIOM_1.1); 1519 (REMIND-MAgPIE 2.1-4.2)  (Harmsen et al., 2021) |
| *Cereal Yield (t DM/ha/yr.)* | 3.7 | 3.7 (IMAGE3.2); 3.8 (MESSAGEix-GLOBIOM_1.1); 3.5 (REMIND-MAgPIE 2.1-4.2)  (Harmsen et al., 2021) |
| *Yield Sugarcane (t DM/ha/yr.)* | 18.7 | 8.6 (IMAGE3.2); 19.8 (MESSAGEix-GLOBIOM_1.1); 30.6 (REMIND-MAgPIE 2.1-4.2) (Harmsen et al., 2021) |
| *Water Withdrawals (km3/yr.)* | 3656-33659 | 2200 – 4200 (Burek et al., 2020) , 3912 (Sutanudjaja et al., 2018) |
| *Water Resource (km3/yr.)* | 47220 | 51800±1800 (Burek et al., 2020); 42393 (Sutanudjaja et al., 2018) ; 42000 – 66000 (Haddeland et al., 2014) |
| *Groundwater Recharge (km3/yr.)* | 15000 | 19000 920 (Burek et al., 2020); 27756; 12666 – 29 900 (Mohan et al. 2018) |
| *Agriculture Withdrawal (km3/yr.)* | 2666 | 2000 [1250-2400] (Burek et al., 2020) ;2735 (Sutanudjaja et al., 2018) |
| *Wastewater Collection (km3/yr.)* | 310 | 224.4–226.9 km3 /yr  (Jones et al., 2021)380 km3/yr. (Qadir et al., 2020) |
| *Wastewater Treatment (km3/yr.)* | 155 - 180 km2/yr. | 186.6 km3/yr. – 189 km3/yr (Jones et al., 2021) |

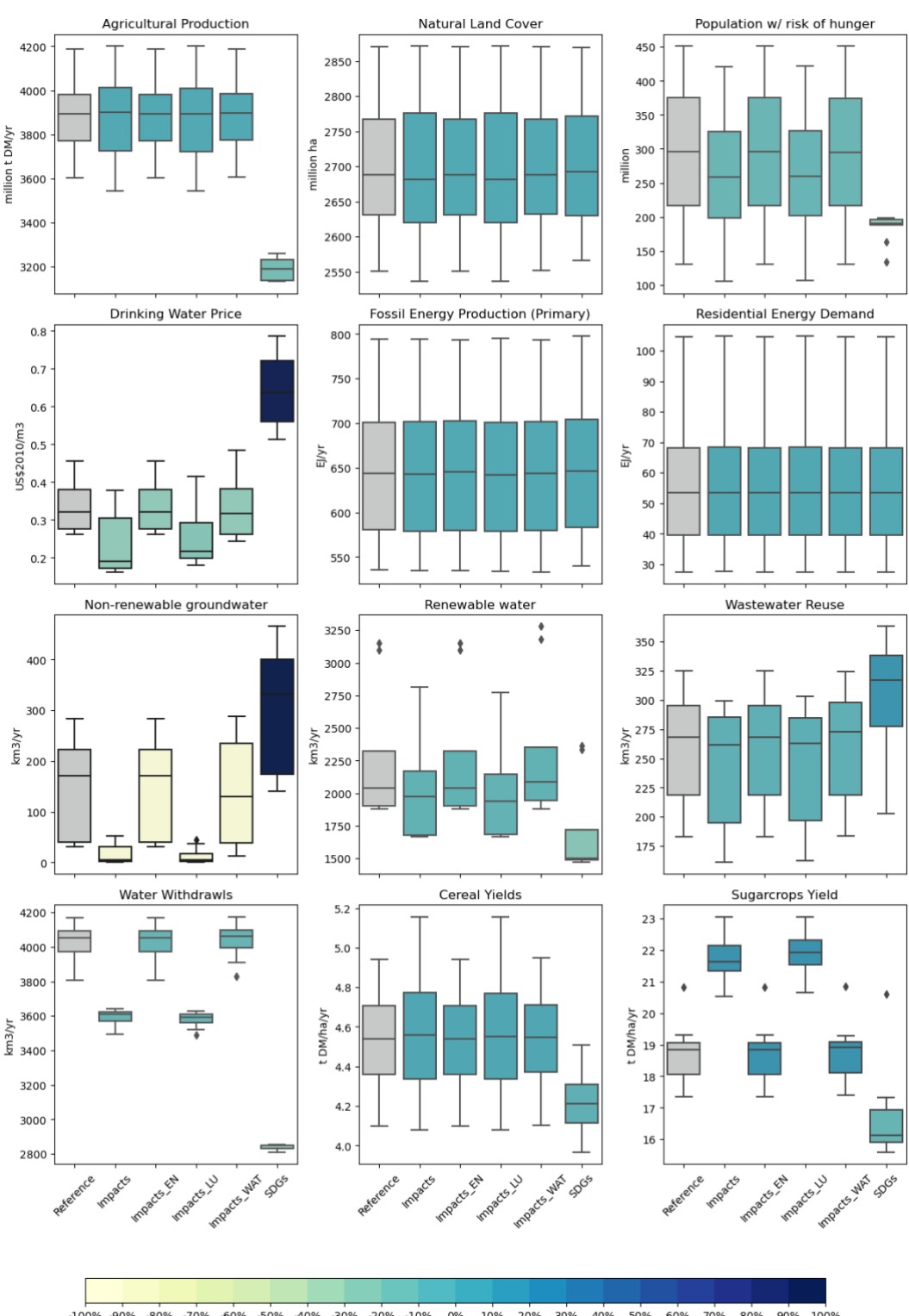

**Figure 4** A Comparison of Key EWL Indicators across Multiple Scenarios It shows the boxplot distributions for selected indicators from the module output. From 2030 to 2080, these are displayed against five distinct scenarios: reference, impacts, impacts_LU, impacts_EN, impacts_WAT, and SDGs. The reference scenario, which stands out visually by having a grey hue, serves as a benchmark for other scenarios. The variance in colour between the remaining boxplots represents the percentage change from the reference scenario.

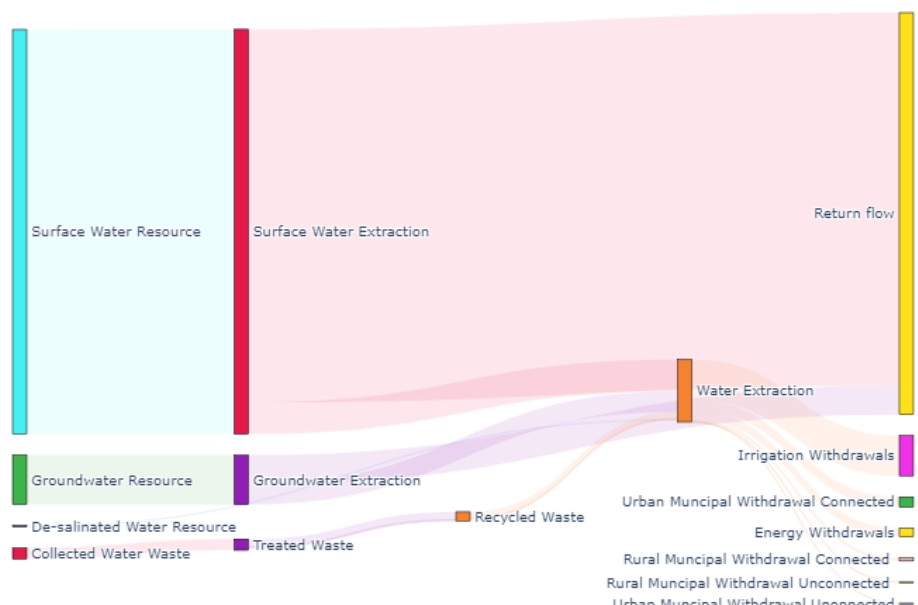

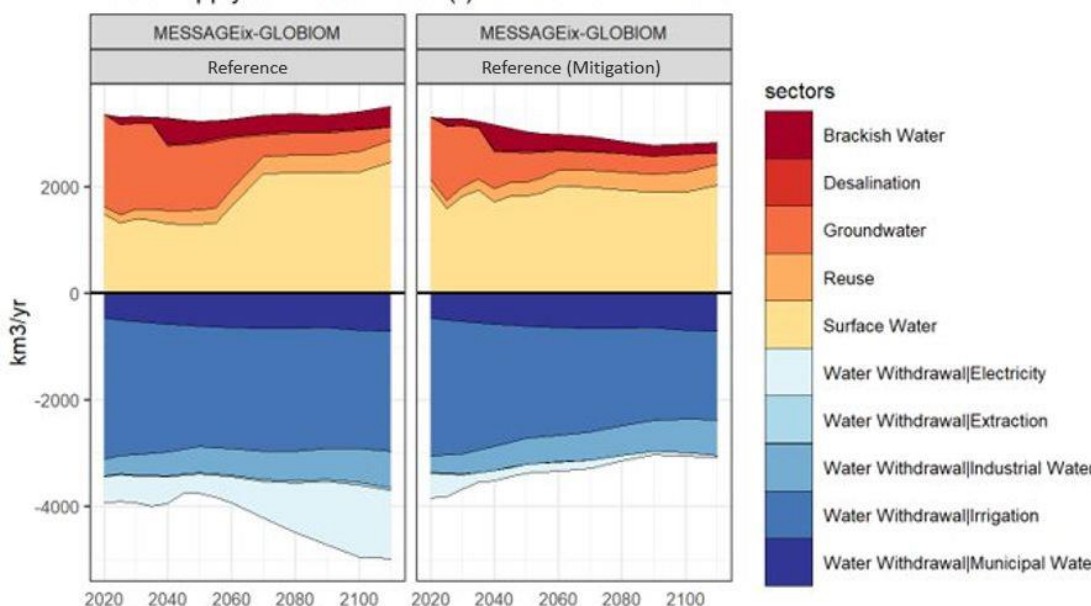

**Figure 5 a)** Water flows from supply to source in the water sector of the MESSAGEix-GLOBIOM nexus module. The flows and associated techno-economic parameters can be tracked as module outputs across the time horizon and scenarios. **b)** illustrates the supply and withdrawal components of the global water balance, which are reported from the module outputs for the Reference and Impact scenarios. A range of blue hues are used to represent the supply sources, and a range of red hues are used to represent the withdrawals.

**5 Discussion**

The MESSAGEix-GLOBIOM nexus module generates outputs that enhance our understanding of the complex interconnections of water, energy, and land, spanning from specific basins to the global scale. The outputs include assessments of water availability, indicators for Sustainable Development Goals, and climate impacts unique to different sectors. These outputs serve as the foundation for conducting integrated route analysis. Figure 6 provides a concise representation of the various outputs that can be generated by the module, emphasizing its ability to provide a wide range of scenario combinations. These combinations reveal the fundamental sensitivities and assumptions of various approaches, enabling us to identify effective methods that are adaptable to change and meet the needs of stakeholders.

In order to determine the effectiveness of the module in different climate and SDG scenarios, we developed a set of scenarios based on different assumptions. While theoretically impractical, the Reference scenario acts as a benchmark for determining the outcomes of biophysical impacts by extrapolating previous climatic data into the future. The module also provides crucial investment and capacity projections at five-year intervals, offering insights into the future of water management. In addition, we have compared these indicators to the available literature in Table 5, confirming the dependability of our findings. This research provides a thorough understanding of global water, energy, and land interconnections. It has the potential to influence policy and investment choices, guiding us towards the sustainable use of resources.

Our module effectively addresses the ever-changing climate system by utilizing a combination of internal and external outputs. As an example, we utilize EPIC to gain valuable information about how irrigation affects crop yields. These findings contribute to GLOBIOM, a system that adjusts land use allocations based on the impacts of climate change. The reallocations, namely in the utilization of land for irrigation, contribute to the balancing of water supplies in MESSAGEix-GLOBIOM. This balancing takes into consideration the requirements of different sectors in the face of changing climate conditions.

The reactions in the water sector are determined by the availability of resources. Climate change-induced changes in the water cycle determine how resources are distributed and require finding alternative sources. The energy industry is subject to similar levels of dynamism, as climate changes have an impact on the efficiency of thermal power plants and the feasibility of hydropower projects while also increasing the demand for cooling. Our module provides a comprehensive multi-sectoral assessment by taking into account these biophysical consequences.

Understanding the interconnectedness of climate impacts across all sectors is essential; the ripple effects they cause require a comprehensive perspective. The results of our study emphasize the need for additional research to fully understand the range of potential effects that climate change could have on many industries and how their inclusion could greatly influence scenarios for managing and reducing these effects. The adoption of sustainable energy sources in certain areas demonstrates the wider significance of our research, which reveals the

interaction between climate effects and strategies for reducing them, along with their additional benefits, such as improved agricultural output and a transition from fossil fuels in the power industry.

Our forthcoming research will expand on these preliminary findings, offering insights that are pertinent to policy-making. The next articles will explore the integration of Sustainable Development Goals (SDGs) with climate policies, providing a fresh outlook on how to tackle climate adaptation problems effectively. While previous research has included SDG components in IAMs, our approach stands out by simultaneously analyzing SDG policies, climate targets, and impacts. This provides a new perspective on the climate adaptation narrative. We utilize this novel methodology to analyze the regional discrepancies in development objectives, facilitating our comprehension of how diverse regions might effectively manage the consequences of climate change while attaining their development targets. The study's regional insights will enhance our understanding of the adaptive strategies that regions may employ to achieve their developmental goals.

To summarize, the outputs of our module connected to the Sustainable Development Goals (SDGs) have the potential to greatly transform our understanding of human development indicators on a global and regional scale. Through the examination of metrics and the comparison of scenarios with and without SDGs, as illustrated in Figure 7, we emphasize the novelty of integrating SDG scenarios with climate effect evaluations. This comprehensive scenario will support future studies, allowing us to assess the combined effects of actions to reduce and adapt to climate change to achieve sustainable development goals.

| Energy | Water | Land | Socio-economics |
|---|---|---|---|
| • Energy use (Primary, Secondary, Final)<br>• Energy prices<br>• $CO_2$ emissions pathways<br>• Capacity requirements<br>• Energy supply portfolio<br>• A/C cooling gap<br>• Investment pathways<br>• Energy use of water commodities)<br>• Adjusted residential demands with increased access to electricity | • Water withdrawals based on constraints<br>• Water supply outlook (combination of different sources)<br>• Capacity requirements of of water infrastructure technologies (wastewater, water distribution)<br>• Investment in the water infrastructure sector<br>• Drinking & irrigation water marginal prices<br>• Water footprint of energy sector | • Water Withdrawals for Irrigation<br>• Crop Yields<br>• Land Cover (different categories)<br>• Agriculture production & demand<br>• Fertilizer use & intensity<br>• Land use CO2 emissions | • Population with access to drinking water, sanitation<br>• Urban & rural municipal demands<br>• Population with access to electricity, clean-cooking fuels<br>• Population with risk of hunger |

**Figure 6** Summary of output indicators that are possible from the MESSAGEix-GLOBIOM nexus module. These outputs are long term pathways and much of these outputs can be further disaggregated onto the technology level.

### 5.1 Further development

While the module includes detailed implementation of the water sector and representation of biophysical climate impacts, we identify areas where our module lacks certain aspects and uncertainties. Since we look at the integrated systems, we do not include inter-basin or spatial unit transfers, which can be crucial for answering transboundary challenges in the river basins. Moreover, we currently do not account for water storage, a potentially important aspect of

water resource management where we can see the water storage during a high flow season and its use during a low flow season. We use the flow percentiles approach to partially address this concern.

    While the Nexus module employs the robust outputs of the ISIMIP for depicting climate impacts, there are certain challenges from the current set of outputs that are not fully consistent with the input climate scenario assumptions. As soon as updated and aligned ISIMIP outputs

become available, we will conduct a new model run to enhance consistency and reduce uncertainty in our analysis. In addition, the sensitivity of indicators to these impacts and the uncertainty of the Global Hydrological Model (GHM) are more significant than those of climate models. The module's representation of alternative water constraints, such as the economic consequences of fossil groundwater extraction to reduce water consumption, will be explored in

future research by focusing on more realistic groundwater assumptions. The current module structure, which assumes an endogenous adaptation response, may not fully capture the complex dynamics, such as the feedback mechanisms between water availability and energy production, socioeconomic impacts of water scarcity on land use, and long-term societal adaptations to water

stress within the EWL sectors. Future research will focus on integrating these inter-sectoral feedback and dynamic responses to enhance the module's accuracy in depicting the intricacies of the EWL nexus.

    In future research, we plan to expand our exploration of climate impact dimensions to include a more robust handling of statistical climate extremes, aiming for greater resilience in our

module's performance at sub-annual temporal resolutions. Future versions of the module will integrate up-to-date climate impact data and strive for more consistent data sources across sectors.

    In addition, we aim to distinguish the roles of impacts and adaptation responses within the EWL sectors, which will allow for a better understanding of

the role of climate and the responses triggered by these impacts in the models. This future work will contribute to the module's refinement and expansion, resulting in a more comprehensive and accurate representation of the intricate interplay between climate impacts, water policy, and reliability.

### 6 Conclusion

This study addresses the research gap of improved EWL nexus, including biophysical climate impact representation within IAMs, by developing a nexus module for the global

MESSAGEix-GLOBIOM integrated assessment model. It enhances the MESSAGEix framework to study the responses to biophysical climate impacts and water constraints across different scales. Representation of interactions with the water sector has been enhanced by implementing endogenous water sector spatial resolution and water constraints by balancing supply and demand at basin scales globally. It can address nexus synergies and trade-offs across EWL sectors on a global scale, showing regional results.

Moreover, the study shows that regional differences influence the cost of alternate water sources and infrastructure. Furthermore, the research on climate impacts highlights the biophysical consequences of climate change on many sectors and the necessity for additional research to comprehend their prospective outcomes. The study also investigates the effects of climate change on the power generation mix, highlighting the transition from fossil to renewable technologies. The results suggest that integrating biophysical repercussions can considerably impact the outcomes of climatic scenarios, and these findings should be regarded in the context of the entire model.

The module is improved to implement river ecosystem constraints, increasing socioeconomic demands, and ecological uncertainties. The module is developed consistent with state-of-the-art software development practices. The whole framework is transparent and flexible to be downscaled to any basin or country worldwide. A first order module can be rapidly prototyped and further used to answer cutting-edge policy questions on the impacts and adaptation potentials across different basins, utilizing a set of socioeconomic and climate ensemble scenarios. The research will address the EWL nexus dynamics and interactions in terms of costs and structural changes concerning future resilient pathways.

**Author's Contributions**

MA, AV, EB, VK, KR conceived the modelling framework. MA & AV led the model development with support of EB, OF, PNK, KR, and VK. SF, EB, AM helped with the GLOBIOM scenarios. AM provided data on cooling gaps, MPC ran energy access scenarios, PB and YS provided the hydrological data. MM, KR, VK did overall supervision. MA led the manuscript, and preparation of results. AV & EB coordinated the overall research. All authors reviewed and contributed to the manuscript writing.

**Data Availability**

The code, processed data, and documentation are available at https://doi.org/10.5281/zenodo.7687578

**Acknowledgments**

We thank Michelle van Vliet for providing the data for climate impacts on cooling technology and Michaela Werning for helping with spatial processing for the desalination data.

We also acknowledge funding provided by the NAVIGATE project (H2020/2019-2023, grant agreement number 821124) of the European Commission.

**Competing Interests**

The authors declare that they have no conflict of interest.

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
