# Peer review of "MESSAGEix-GLOBIOM Nexus Module: Integrating water sector and climate impacts"

_EGUsphere, 2023_

## Author Comment (AC1)

**Reviewer 1**

**This paper describes the advancement of the MESSAGEix-GLOBIUM integrated assessment modeling framework through the integration of a new module, the nexus module. This module provides a detailed representation of the water sector with high spatial (basin level) resolution and describes the interaction of the water with the energy and land sector. This is a crucial component for a comprehensive evaluation of the sectoral interaction in the face of climate policy measures, climate change impacts and sustainable development goals, as outlined by the paper. Therefore, this constitutes an important step to allow the model to be used in highly policy relevant applications and I find this contribution important and fitting to GMD.**

**The paper describes the new module as well as exemplary applications, in particular an analysis of effects of climate change impacts in two policy scenarios. It However, it refrains from detailed analysis and interpretation of results, referring to planned future publications, thereby remaining quite vague. While the paper is well structured and gives a reasonable overview of the modeling approach. However, it would benefit from a critical revision regarding clarity and language. I go into details on this in the next section.**

Thank you for your insightful comments and favorable evaluation of our manuscript. We appreciate your acknowledgement of the significance of our work.

We acknowledge your comment regarding the limited analysis and interpretation of results in the current manuscript. In response to your comments, we have revised the manuscript and have restructured specifically the results and discussion to provide more meaningful insights and also validates our work with existing literature studies.

We have also thoroughly reviewed the manuscript with a focus on enhancing the overall readability and ensuring that our modeling approach is presented in a clear and concise manner.

We believe that these revisions have substantially improved the manuscript, resulting in a more robust and informative description of our modeling framework and its applications. The original comments are in **bold,** our comments are in standard text, and the additions/revisions are indicated in *Italics.*

**Specific comments**

> **RC1.1.     I think the clarity of the description could be improved in two ways. The authors seem to assume a certain level of knowledge from the readers about the MESSAGEix-GLOBIOM model and its applications. It is not clearly explained what the core application of the model is and how the new module extends that. In particular, it does not become clear what is assumed along given scenarios and what is subject to endogenous optimization decisions for readers without much previous knowledge. The term "optimization" is mentioned in line 137 without further explanation. Similarly, what is new regarding the representation of climate impacts and SDG constraints, and what has been applied before (e.g., on the land use and energy side)?**
>
> We appreciate your feedback. We concur that the clarity of the description could be enhanced, particularly with regard to the explanation of the MESSAGEix-GLOBIOM model's core application and how the new module extends it. In the revised version, we have provided a more comprehensive description of the model's core application and how the new module expands its capabilities. We have also highlighted the endogenous optimization decisions within the model and clarified the assumptions made along given scenarios.
>
> In addition, we have addressed the mention of "optimization" in line 137 by providing a more detailed explanation of the model's optimization process. In addition, we have elaborated on what the innovation of this study is. The subsequent paragraph reflects these modifications:

*The "nexus" module of the MESSAGEix-GLOBIOM framework, MESSAGEix-GLOBIOM Nexus v1 presented in this paper, contains endogenous spatially- and temporally explicit climate impact constraints and water allocation algorithms. This module extends the foundational work carried out by* (Parkinson et al., 2019). *It addresses the gaps in the previous study by improving the water sector resolution, water constraints, and climate impacts. The module here refers to expanding the core global framework to represent specific dimensions straightforwardly at the cost of increased computational complexity and cost. The MESSAGEix-GLOBIOM Integrated Assessment framework is a global energy-economic-agricultural-land use model that evaluates the interconnected global energy systems, agriculture, land use, climate, and the economy. Using a linear programming approach, the MESSAGEix framework optimizes the total discounted system costs across all energy, land-use, and water sector representations. It provides options for both perfect foresight and recursive-dynamic modes. Its adaptability and flexibility make it a powerful instrument for optimizing transformation pathways at various scales, emphasizing minimizing system costs. It comprises five complementary models or modules: the energy model MESSAGEix* (Huppmann et al., 2019), *the land use model GLOBIOM* (Havlík et al., 2014), *the air pollution and greenhouse gas (GHG) model GAINS, the aggregated macro-economic model MACRO, and the simple climate model MAGICC* (Meinshausen et al., 2011). *The framework combines the MESSAGEix and GLOBIOM models to assess and model policy scenarios' economic, social, and environmental implications. The framework comprehensively examines the trade-offs and synergies between numerous policy objectives, such as reducing greenhouse gas emissions, boosting food security, and safeguarding natural resources. To access sustainable development targets, the framework is utilized to evaluate the feasibility and implications of alternative policy choices and to guide decision-making.*

**RC1.2.     Similarly, what is new regarding the representation of climate impacts and SDG constraints, and what has been applied before (e.g., on the land use and energy side)?**

We agree that mentioning previous literature and what have been new from the previous versions of the MESSAGEix-GLOBIOM could help the readers understand the novelty of this work. We have incorporated these changes in the modified document. The revised paragraph mentioned under RC1.1 reflects this concern and also the paragraphs added as response to RC1.5 further clarifies the novelty of representation.

**RC1.3.     The introduction would benefit from some streamlining and more precise wording and description. I refer to some examples of unclear sentences in the technical section below. It would be helpful to clearly distinguish between underlying scenarios (the SSPs, which are combined with the RCPs in climate policy analysis), how models implement them, and what models analyze additionally with those scenarios as a backdrop (e.g., transformations to achieve the given climate policy goals).**

We have added a sentence to explain it in the introduction of the model:

*The module uses SSP-RCP (Shared Socioeconomic Pathways – Representative Concentration Pathway) combinations as narratives for creating a baseline scenario. Each scenario is developed using SSP-RCP combinations, national policies, and Sustainable Development Goal (SDG) assumptions aggregated at the R11 region. These SSP-RCP combinations help to formulate climate impacts, adaptation needs, and mitigation scenarios under different socioeconomic assumptions.*

We have added a further explanation of the scenario used for the current study in the Results & Discussion section too:

*As mentioned in the previous section, any scenario combinations are possible from the module. However, to test the model's applicability across climate and SDG scenarios in combination, we formulated a total of six scenarios that alternate different assumptions. For the current setup, we used a combination of SSP2 pathways and combined these with RCP2.6 & RCP6.0. The upcoming work will include more SSP dimensions in combination with RCPs to have more consistent assumptions across scenarios.*

*The scenario formulation we used to describe the results is.*

- **Reference** *scenario includes historical climate assumptions. The data used in this scenario doesn't include any climate effects for the future.*
- **The impacts** *scenario includes climate impacts across the EWL sectors. This scenario assumes the RCP6.0 scenario for different biophysical climate impact indicators, as indicated in section 3.2.*
- **Impacts_LU** *scenario assumes only land use impacts from GLOBIOM.*
- **Impacts_WAT** *scenario assumes only water sector impacts on the renewable water availability and capacity factors of cooling technologies for thermal power plants.*
- **Impacts_EN** *scenario assumes the energy sector impacts, including the hydropower impacts and cooling/heating energy demands.*
- **SDGs** *include all SDG-related assumptions indicated in section 3.4*

**RC1.4.** **The introduction currently fails to embed the new advances properly in the IAM literature, e.g., distinguishing process-based from cost-benefit IAMs, describing which (if any) other process-based IAMs capture climate impacts/SDGs/the water sector and how this implementation differs from other approaches. The citations in the first paragraph are relevant but also quite dated. In line 52, it is mentioned that "Many IAMs consider adaptation costs in an aggregated spatial region" – I think most IAMs do not consider adaptation costs (aside from pretty old approaches); if there are examples, please provide some references.**

Thank you for your comments. We realized the lack of specificity in describing the advancements in the IAM literature and for the out-of-date references. To address these concerns, we have made significant modifications in the introduction section to incorporate the most recent developments relevant to this study. Regarding the mention of adaptation costs in line 52, We have removed the sentence claiming that the majority of IAMs do not take adaptation costs into account.

The improved text addressing the reviewer's concern is:

*Impact modeling activities across diverse modeling groups, such as the Intercomparison Model project (ISIMIP)* (Frieler et al., 2017)*, have been carried out to understand the impacts of climate change better individually. These sectoral exercises include assessments of changing yields, runoff changes, food production, and groundwater. estimate that economic impacts have been estimated using a variety of methodologies, depending on the types of impacts considered, such as the relationship between climate damages and temperature. Some studies have empirically linked climate conditions with socioeconomic systems and incorporated distributional factors into cost-benefit models, resulting in increased social costs of carbon and more stringent mitigation pathways* (Dellink et al., 2019; Diaz and Moore, 2017; Moore and Diaz, 2015; Kalkuhl and Wenz, 2020). *It is becoming quite evident to have the representation of biophysical climate impacts into integrated assessment models to comprehend the effects of different sectors on the techno-economic outlook and to determine mitigation and adaptation pathways.* (Piontek et al., 2021) *analyzed the economic impacts of climate change using the REMIND IAM model, but biophysical*

*climate impacts were not represented.*(Soergel et al., 2021a, b) *emphasized the significance of considering the consequences of climate impacts and evaluating how integrated scenarios respond to these impacts, especially regarding sustainable development pathways. This study addresses these gaps by proposing a framework that integrates climate impacts, strengthens the water sector (which is essential in the context of climate change), and formulates scenarios in conjunction with sustainable development assumptions to assess the impacts of climate change under mitigation, adaptation, and sustainable development pathways.*

**RC1.5.** **In the introduction and also in the discussion, the possible application in the context of evaluating adaptation measures is emphasized. However, it does not become clear how that could be done, i.e., just through endogenous model responses to climate impacts like increased irrigation, land-use change, or explicit adaptation policy. The reader expects more (technical/methodological) detail from the introduction, even if an actual application is planned in a future paper.**

We agree that this aspect was missing in the previous version. We have added some lines in the revised text clarifying the potential applications of the module in the revised manuscript in the following way:

*This paper introduces a new module of the global MESSAGEix-GLOBIOM framework* (Riahi et al., 2021; Krey et al., 2016). *The nexus module attempts to fill the gap in integrated assessments by improving the representation of biophysical climate impacts across the Energy, Water Land (EWL) sectors and enhancing the water sector representation. Using this module, we develop scenarios that can effectively capture climate impacts across multiple sectors. Then these scenarios are combined with SDG targets in EWL sectors to capture the synergies and trade-offs of climate impacts and sustainable development pathways.*

*One of the critical features of the Nexus module is its ability to simulate global interactions across multiple sectors and systems. It allows the model to represent the complex feedback and spillover effects from policy interventions, such as the potential implications of land use changes on the global food system and the energy sector or the water footprints of the energy system. The framework allows a realistic and complete study of policy possibilities by incorporating many facts and hypotheses, such as population and economic growth predictions, technology advancement, and resource restrictions. The integrated approach thoroughly considers the trade-offs and synergies between diverse policy objectives, such as reducing greenhouse gas emissions, enhancing food security, and protecting natural resources. Considering biophysical climate impacts across different sectors helps to access different adaptation needs and responses in different sectoral outputs across different pathways. In the context of sustainable development, it can analyze the viability and implications of various policy alternatives and inform decision-making.*

We have also added the explanation on the responses of climate impacts within the Discussion section on how these impacts helps allocate decision in an integrated manner:

*To capture the dynamic responses of the climate system, the model's response to climate impacts employs a multifaceted strategy that includes both endogenized and exogenous outputs. The use of the EPIC model, which provides information on irrigation responses and their subsequent effects on crop yields, is one prominent example. Then, these yield outputs are incorporated into the GLOBIOM model, where adaptation responses are endogenized, causing a reallocation of land use system resources based on climate impacts. Notably, this reallocation includes decisions regarding land use that directly affect water use in irrigation. The irrigation withdrawal computations are then used by the MESSAGEix GLOBIOM model, which effectively balances water supplies by considering irrigation withdrawals in conjunction with withdrawals from other sectors under changing climate conditions. In contrast, responses in the water sector are contingent on the availability of resources under various climate scenarios. The effects of climate change on hydrology have a direct impact on the availability of resources, compelling the model to adapt and consider alternative supply sources. Similarly, the energy sector incorporates endogenized decisions based on the effects of climate-induced changes in the capacity factor of thermal power plants. These changes have implications for thermal power generation and the feasibility of hydropower installations in various regions. Additionally, the demand for cooling is acknowledged as a significant factor influencing energy demands. Through this integrated approach, the model systematically accounts for and responds to the biophysical impacts induced by a changing climate, providing a comprehensive assessment of the interdependence and implications across multiple sectors.*

**RC1.6.    There are a few things that do not become clear regarding the implementation and would benefit from an improved description. Here are some examples. The introduction (page 2) describes the need for higher spatial resolution of water. However, that is also true for land (and handled like this in the EPIC-GLOBIOM connection) – maybe worth pointing this out?**

We have clarified further in the revised manuscript and improved the explanation in the following manner:

*Land use dynamics are modeled using an emulator that links the GLOBIOM model* (Frank et al., 2021) *to the Global Forest Model (G4M)* (Gusti, 2010). *The GLOBIOM is a global recursive dynamic partial equilibrium model of the forest and agricultural sectors. It uses a bottom-up approach and simulation Units (SimU) to model agricultural and forest productivity* (Frank et al., 2021). *The emulator integrates a set of land-use scenarios, so-called lookup tables, comprised of different biomass- and land-use emission potentials. The land-use scenarios inform the energy model of biomass availability at given price levels and, in addition to that, associated GHG emissions from the land-use sector. Each land-use scenario is complemented by several other indicators, for example, land-cover developments. Irrigation water withdrawals, an indicator also tracked as part of the GLOBIOM emulator, are endogenized in the nexus module as irrigation water demand. The annual irrigation water requirements across different scenarios are simulated using the EPIC biophysical crop model*(Balkovič et al., 2014) *at a 0.5° x 0.5°*

*spatial resolution, distributed monthly over the growing season based on local cropping calendars for a 10-year time step. These requirements are used as input to the GLOBIOM model. The GLOBIOM model upscales these water requirements and provides irrigation requirements at an aggregated 37 regions based on land-use allocation decisions.*

**RC1.7.** **Further, there is the issue of cooling water for thermal power plants. That depends on water temperature but does not use water up without feeding it back into rivers, to my knowledge. Is that how it is considered here, and where does the input data on water temperature come from?**

We used data by (Yalew et al., 2020) to include the capacity factors of cooling technologies power plants. (Yalew et al., 2020) calculated these capacity factors as a function of different climate-dependent assumptions, including temperature. The capacity factors impact the quantity of water required for cooling of thermal plant and also the fossil fueled generation ultimately. Furthermore, we account for withdrawals and consumption of cooling water discharge for thermal power plants however in our model, those are note temperature dependent.  In order to clarify, how water withdrawals and consumption factors are included, we have added the following text in the revised manuscript:

*For calculating water withdrawal and return flows from energy technologies, we use the approach detailed by (Fricko et al., 2016).Each energy technology requiring water is provided with a withdrawal and consumption intensity (e.g., cubic kilometers per GWh), which then allows the model to translate technology outputs into water requirements and return flows, which in turn balance with the available supply. For power plant cooling technologies, where the water requirements are calculated as a function of heat rate, the efficiency change in the energy technologies (e.g., lower heat rates) impacts the cooling requirements per unit of electricity produced. The withdrawal and consumption intensities for power plant cooling technologies align with the range reported by (Meldrum et al., 2013), while additional electricity demands from recirculating and dry cooling technologies are included in the electricity balance computation. Other technologies adhere to the data in (Fricko et al., 2016).*

**RC1.8.** **In Table 1, more information on critical assumptions behind some input data would be helpful here (without having to look them up in all the other papers), e.g., one wonders how future water demand data are projected into the future.**

Table 1 is meant to summarize different data assumptions and references used in the water sector, along with pointing them out to the folder in the data repository shared with the paper submission. We have added an explanation of water demands in section 3.1.

*For calculating the municipal water demands, we used the approach followed by (Graham et al., 2020). Urban and rural components of municipal water demand projections are calculated using gridded population and income-level projections data by (Wang and Sun, 2022). Manufacturing demands are generated following a similar approach used by (Hejazi et al., 2014). Historical country-level data for 2015 is estimated by subtracting energy sector withdrawals from total industrial sector withdrawals. Future changes in manufacturing demands are projected, assuming convergence towards a log-linear model between GDP and manufacturing withdrawals. Demands are distributed across countries based on growth in GDP and then downscaled to 7.5 arcminutes and re-aggregated at the B210 basins. Supplementary Figure S shows urban and rural components of municipal demands and industrial demands for 2050, whereas the data is provided in the GitHub repository (See Data Availability). Supplementary Figures S3.1 & S3.2 shows average municipal and industrial demands across the basins.*

**RC1.9.** **In the first paragraph on page 11, it is mentioned that an iteration between MESSAGE and the energy access model is not done because it would alter the GDP pathways. How much would they be altered?**

As indicated in the paper, we have done one iteration with the MESSAGEix-GLOBIOM nexus to reducing computational complexity and alteration of GDP pathways. However, (Poblete-Cazenave et al., 2021) mention some regional examples in Table A3 for Ghana, India, South Africa, and Guatemala. Overall, we see the GDP increase of more than 100% from the baseline levels (no energy access), with India being the highest.

**RC1.10.** **In the discussion of the environmental flow constraint, a categorization of basins based on development status is mentioned (line 321) – is that static or changing in the future? In line 329, there is a distinction between basins with high and low adaptive capacity – do the ones with low capacity never have any targets or just later?**

We divided basins with high adaptive capacity or low adaptive capacity using the current indicators of development status from World Bank (https://databank.worldbank.org/source/world-development-indicators). The basins with low adaptive capacity or low development status implement environmental flow constraints from 2040 onwards instead of 2030 for the high adaptive capacity. We have adjusted the relevant text in the revised article.

*We categorized these basins based on the development status of countries specified by the World Bank and implemented a lower environmental flow target in the respective regions from 2030 onwards and increased the target till 2050, thus following the trajectory of basins with high adaptive capacity.*

**RC1.11.** **How is groundwater treated in regions which already overuse it today?**

We use simplified assumptions for groundwater as a water supply source in the model. We consider the groundwater recharge from the CWatM as renewable groundwater and use historical surface-groundwater ratios to calibrate the future ratio of groundwater usage. Non-sustainable groundwater abstraction is motivated from the demand side and considered infinite in the model. Future work will improve non-sustainable groundwater extraction.

**RC1.12.** **The results section is very generic in its discussion, so it is not very helpful and leaves the interested reader with many questions. Maybe a detailed discussion of one application, plus a generic list of possible future research questions to be answered with this framework, would be better. It needs to be explained better what flow reliability in section 4 means, and this application needs to be motivated. In the application of impacts and the energy sector, the mention of mitigation is missing. How do the impacts interact with that? Are there any impacts on bioenergy, for example? Is the lower use of fossil energy primarily due to the cooling water issue, or what other drivers are there? Why are the impacts in some regions like Central Asia larger under RCP2.6 than 6.0? In the water extraction example, brackish water appears for the first time. From the figure, its use seems to go down, but in the text, a rise in its usage is mentioned. In the first paragraph on page 15, regional effects are discussed, but regions are not named ("certain regions" in line 445). The discussion there does not seem of the results but is very generic (line 452: "impacts of climate change on water availability are likely to be negative" – the authors know what they put in from the biophysical modeling, right?). In line 458, the topic suddenly switches to yields.**

Thank you for your feedback regarding the Results and Discussion section. We value your input and have made modifications based on the suggestions you provided. We have provided more specific and detailed information and discussed the interaction between impacts and sustainable development targets. In addition to focusing on the effects of climate change on water availability, we have provided a more summarized analysis of the key indicators across different scenarios. We have also performed new model runs and formulated new scenarios to better frame our results section. We believe these modifications improve the clarity and relevance of the section on the results. We have added the detailed modification under the Reviewer Comment (RC 2.15).

**RC1.13.** **Finally, the paper misses a critical discussion of assumptions. For example, why is the entire ensemble of ISIMIP water models used for desalination potential but not for water availability? What about uncertainty from impact models? Is the uncertainty from climate models only used for water availability or other parts like yield changes?**

We appreciate the reviewer's insightful comments on our paper's discussion of assumptions. In accordance with the ISIMIP 3b protocol, water-related outputs were generated using the preferred climate model GFDL-ESM4 as specified in their technical documentation. However, we acknowledge that utilizing the entire ensemble of ISIMIP water models for desalination potential, as was done for water availability, would have provided a comprehensive assessment of model uncertainty.

Regarding climate model uncertainty, it is essential to note that it was primarily evaluated in terms of water availability rather than yield changes or other factors. This decision was based on our study's specific objectives and focus. However, we acknowledge that incorporating uncertainty from climate models into other components, such as yield changes, could result in a more thorough analysis of potential variations.

Theoretically, it is possible to explore different climate model results and evaluate their effects on both land and hydrological inputs for the MESSAGE model. However, implementing such an approach involves rerunning both the GLOBIOM and MESSAGE models. This can be a complex and computationally intensive task in practice. Before selecting the ensemble for our analysis, we investigated the outcomes of a variety of climate models. This procedure involved evaluating the outputs of numerous climate models to ensure a representative and accurate representation of the spectrum of possible climate scenarios.

**RC1.14.     Minor technical points/clarifications:**

We appreciate the reviewer for indicating minor clarifications and technical points. We have added the revised sentences below each minor technical point/clarification point and updated in the revised manuscript.

**Line 32/33: "… pose an extra threat to climate change risk" – maybe "additional to climate change risk"?**

*In addition to climate change risks, limited resources compounded by population and GDP growth pose an additional challenge.*(Byers et al., 2018)

**Line 38: "Transition to ambitious global warming goals" – better "ambitious climate policy goals"**

*IAMs provide long-term transformation pathways to answer critical questions on climate change transition to ambitious climate policy goals.*

**Line 50: "Due to substantial challenges in technical implementation and representation" – representation of what?**

*Although SSPs were designed to analyze the challenges for mitigation and climate adaptation, integration of climate impacts and adaptation of energy and land sectors to water sector constraints has, until recently, been relatively limited in the IAM scenarios* (Riahi et al., 2017b) *due to substantial challenges in technical implementation and representation of climate impacts.*

**Line 59: "Need for a balanced integration" – what do you mean by balanced?**

*There is a need for a balanced synthesis of Shared Socioeconomic Pathways (SSP) narratives with climate impacts, adaptation, and resilience pathways to assess water, food, and energy security to access sectoral adaptation costs and impacts.*

**Line 65: You mention community resilience, but that isn't resolved in the model.**

*The integration of cross-sectoral Energy Water Land nexus analysis in IAMs can help identify trade-offs and synergies, integrate policy implementations, and address equity dimensions, such as the population exposed to hunger or lacking access to sanitation and electricity. This holistic approach enhances the resilience of communities and promotes sustainable development.*

**Line 114: "It simultaneously determines energy, land use." – what do you mean by energy – energy demand, supply, the mix of energy sources...?**

*The nexus module simultaneously determines energy portfolio, land use, and associated water requirements, and feedback from constrained resources, such as limited water availability for energy and land use resource usage.*

**Line 116: The acronym GHM is not explained**.

The acronym Global Hydrological Model (GHM) is added.

**Line 165: "Using a set of configurations in the energy system" – what do you mean by that?**

*The scenario is further extended from the typical scenario in the nexus module using certain policy and technological assumptions.*

**Caption Figure 2: typo: "The water system is mode*led*"**

The typing error is fixed.

**Line 207: the word "driving" seems not to fit**

*Three bias-corrected meteorological forcing data outputs from different climate models (GFDL-ESM2M, HadGEM2-ES, IPSL-CM5A-LR) are used to estimate surface runoff and groundwater recharge.*

**Line 252: I assume you refer to Figure 2 in the Gernaat paper – maybe make that clear?**

We removed this part to avoid confusion.

**Line 316: The sentence "The rivers' environmental flows help protect … from achieving SDG target 6.6" – It is unclear what that means. Also, not all readers will be aware what specific sub-targets of the SDGs are, so they should be explained if mentioned. The sentence after starting with "We use the" is double.**

*The rivers' environmental flows help protect river-related ecosystems from achieving SDG target 6.6 (protect and restore water-related ecosystems, including mountains, forests, wetlands, rivers, aquifers and lakes).*

We removed the duplicate sentence.

**Line 402: Some words seem to be missing.**

We removed this paragraph completely in the revised manuscript.

**Line 487: What is meant by "most be more tolerant of statistical**

*For instance, the sub-annual temporal resolution must be more tolerant of climate extremes under various circumstances.*

**Reviewer 2 – Page Kyle**

RC2.1.    **Overall, the study marks an advancement in integrated assessment modeling, tightening the coupling between biophysical and economic systems and improving the capability to construct internally consistent scenarios. The paper summarizes a lot of good methodological work. Specifically, the features added include climate impacts on agriculture, water, and energy, and an enhanced representation of the water sector. The paper would benefit from clearly articulating what is different in the study and modeling system here as compared with the prior cited model documentation (Krey et al. 2016). As well, most of the cited literature in the introduction is pretty old and doesn't necessarily reflect the state of the art in integrated assessment modeling which has been focused on improving things like hydrology and climate impacts in the past 5 or so years. Most of the references here are from 2013-2017. Since the study addresses the sustainable development goals, a nod to the recent literature on using IAMs to quantify the SDGs would also be appropriate; I've provided some specific examples below.**

We sincerely value the reviewer's comprehensive analysis and insightful comments on our manuscript. Their feedback has helped us improve the study's clarity and relevance. We have carefully addressed each comment and, where necessary, provided clarifications. In addition, we have incorporated the reviewer's recommendation to distinguish our study from previously cited model documentation (Krey et al., 2016). We have also updated our citations to include more recent research, particularly concerning advances in integrated assessment modeling, hydrology, climate impacts, and the quantification of Sustainable Development Goals (SDGs). We appreciate the reviewer's contribution. The revised text addressing these changes is mentioned below the comments RC 1.1,1.3,1.5 & 1.6.

RC2.2.    **The biggest issue that I have with the study is that the results that are shown are not especially meaningful indicators (among the set that could be shown) and are not explained or defended in the discourse of the results and discussion section, which right now is written largely as an extension of the introduction. A significant portion of the results and discussion is invested into background information about combined systems modeling, with commentary about why these modeling capabilities are novel and useful, but there's only sparse comparison between these results and the existing literature on similar indicator variables. Many of the key results are counter-intuitive for me, which is OK, but the problem is that the results aren't explained or defended. The result that most stands out to me here is that some of the highest levels of investment into water infrastructure globally, indicated in billions of dollars per year per water basin, are seen in mostly uninhabited Arctic basins such as Alaska, the Yukon, and Siberia, that generally have an over-abundance of freshwater. The authors clarify in the text that there aren't any inter-basin transfers, which was the only explanation for the result that I could think of (e.g., NAWAPA-type projects). Similarly, almost all of the results that are shown in the key figures (5 and 6) are indicated in units that make the logical comparisons in the figures largely meaningless. For example, Figure 5b is set up to compare the different global macro-regions, but the variable chosen is the total cost of water investment, in billions of dollars per year. The indicator isn't normalized for each region's total water supply, economic output, or population size. So, Eastern Europe sees the lowest investment costs among regions, but this is probably because it is comparatively small, and without that normalization there's not much that can be learned from the figure in terms of comparing the regions. That figure would probably make the most sense to construct as costs per unit of water supplied, but really that depends on what the figure is being used to demonstrate in the analysis. The authors stress that the results should be interpreted with caution (line 479), but what I'd prefer to see is for the figures to provide meaningful indicator variables, and then to compare the key results to the existing literature, where such literature exists. In some cases, like the water investment quantities shown in Figure 5a, there might not be literature on the topic outside of this research team, and that's worth highlighting too. The following are my specific comments:**

We appreciate the reviewer's detailed comment and the provided insights. We have carefully considered the concerns raised and made significant modifications to address them. Specifically, we have reevaluated the scenario selection and regional representation. We now present global results to facilitate comparisons with other sources of literature and to ensure the validity of our findings. In addition, we have reexamined the indicator variables used in our figures to provide comparisons that are more meaningful and interpretable. We recognize that the initial presentation may have lacked sufficient context and justification for the results, and we have revised the results and discussion section

accordingly. We sincerely appreciate the reviewer's meticulous work, which has significantly enhanced the clarity and readability of our study. We have faith that the revised results adequately address the reviewer's concerns. We have added the modified text, including figures and tables, below in response to RC2.15.

**RC2.3.**    **Section 2: in the description here, there is no distinction drawn between withdrawals and consumption. It should be clarified which one is used (seems to be withdrawals), and the representation of return flows should also be described. Power plants using once-through flow cooling systems don't drive water scarcity, though a withdrawals-focused accounting framework could find such a result if return flows aren't tracked.**

Thanks for your valuable feedback. Indeed, we agree that using withdrawal focused results specifically in once through cooling technology systems don't drive water scarcity. Therefore, we account for water consumptions in cooling technologies We have addressed your concern by providing a clearer explanation (mentioned below) in the water section of the revised manuscript:

> *For calculating water withdrawal and return flows from energy technologies, we use the approach detailed by* (Fricko et al., 2016).*Each energy technology requiring water is provided with a withdrawal and consumption intensity (e.g., cubic kilometers per GWh), which then allows the model to translate technology outputs into water requirements and return flows, which in turn balance with the available supply. For power plant cooling technologies, where the water requirements are calculated as a function of heat rate, the efficiency change in the energy technologies (e.g., lower heat rates) impacts the cooling requirements per unit of electricity produced. The withdrawal and consumption intensities for power plant cooling technologies align with the range reported by* (Meldrum et al., 2013)*, while additional electricity demands from recirculating and dry cooling technologies are included in the electricity balance computation. Other technologies adhere to the data in* (Fricko et al., 2016).

**RC2.4.**    **Line 150: Does the model consider drip irrigation or other technologies that could reduce the irrigation water intensity of irrigated crop production? Water use efficiency is introduced as a concept in the discussion (line 459) and would seem to be one of the few ways of simultaneously meeting SDGs 2 and 6, but there isn't any description in the methods.**

We do not include specific irrigation technologies, such as drip irrigation, in our current model structure. Nevertheless, the water uses from the GLOBIOM component of the MESSAGEix-GLOBIOM nexus model assume reduced water withdrawals and an increased allocation of water for environmental flows, which indirectly accounts for increased water use efficiency across different climate and sustainability assumptions (Pastor et al., 2019). We have also further elaborated this in the revised manuscript:

*The irrigation conservation approach is implemented to reduce the irrigation withdrawals and reallocate water to other sectors, thus contributing to target 6.4 (Frank et al., 2021). (Pastor et al., 2019) mentions how the reduced water approach in the irrigation sector in the GLOBIOM model accounts for environmental flows and the water is re-allocated to environment and domestic uses by saving from the irrigation sector.*

**RC2.5.**    **Line 179: "water withdrawals for irrigation, energy, and cooling" - throughout the text, the term "cooling" is used interchangeably for buildings air conditioning and for thermo-electric power plant cooling. Sometimes it isn't clear if both are intended, and large buildings sometimes use water-based "chillers" for air conditioning. The industrial sector also uses water for process cooling and it isn't clear whether that's classified as such here. Please clarify what sectors/processes are intended whenever the term "cooling" is used, if it's not obvious from context.**

We appreciate your clarification comment. In our model, "cooling" refers primarily to the cooling of thermoelectric power plants. While the model does account for industrial water demand, it does not account for building-specific water usage, such as air conditioning or chillers. We have ensured in the revised manual that this context is clarified whenever the term "cooling" is used. We have also added the following text in the revised manuscript of calculating industrial demands:

*Historical country-level data for 2015 is estimated by subtracting energy sector withdrawals from total industrial sector withdrawals. Future changes in manufacturing demands are projected assuming convergence towards a log-linear model between GDP and manufacturing withdrawals. Demands are distributed across countries based on growth in GDP, and then downscaled to 7.5 arc-minutes and re-aggregated at the B210 basins. Supplementary Figure S shows urban and rural components of municipal demands and industrial demands for 2050 whereas the data is provided in the GitHub repository (See Data Availability). Supplementary Figures S3.1 & S3.2 shows average municipal and industrial demands across the basins.*

**RC2.6.** **Section 3.1, general: Please provide a brief commentary on how the basin-region crosswalk is handled. There are basins that supply water to multiple regions; for example, the Nile supplies both SSA and MENA. Are the basins disaggregated in the model, such that there are e.g., basins for Nile-SSA and Nile-MENA? In that case, how is the apportioning done? Does the water supply in Nile-MENA only include renewable water from within Nile-MENA, or do its supplies include the runoff in the river, that wasn't consumed by Nile-SSA? Alternatively, do the relevant portions of each region simply share a single water resource base?**

We acknowledge the importance of balancing computational cost and enhanced regional detail in cross-region mapping. To elucidate, we have included an example in the revised manuscript featuring the Nile River basin, which spans South Africa and the Middle East. The water availability however is spatially aggregated from a grid scale to a spatial unit, so we are not accounting further inflows and outflows from the basins.

*To better understand the spatial distribution and water balance of regions, we can look at the Nile River basin, which extends across South Africa and the Middle East (R11 native regions). Due to the overlapping of these two R11 regions, we come up with two distinct spatial units: Nile-Middle East and Nile-South Africa. Now for Nile-South Africa, using proxy indicators such as basin area and the proportion of available water in each basin, we calculate the proportion of renewable water resources available from the Nile and the total water availability in the South African region. This 'downscaled' value plays a crucial role in the model, allowing us to reconcile the available water supply options with the region's varying water demands.*

**RC2.7.** **Section 3.1, general: Please provide a brief commentary on how the energy used by the water sector is handled. This seems like it should be somewhat complex in the modeling system used, as the energy and water modules are run separately, and Table 1 indicates that water distribution and wastewater treatment energy footprints are used, but I didn't see a description otherwise.**

Upon the indicated reviewer comment, we have added the following commentary on how the energy used by the water sector is handled in the revised version of the manuscript:

*The energy footprints of various components of the water sector, including supply (surface water and groundwater extraction), distribution (urban and rural), and wastewater treatment (treatment, recycling, and re-use), are interconnected with the electricity needs of the energy sector. This connection is established through basin-region mapping, which enables the spatial aggregation of appropriate fractions of electricity requirements to the region (R11) where the water sector's electricity consumption is managed. Table 1 indicated different references used for electricity requirements per unit of water infrastructure activity at different stages.*

**RC2.8.** **Lines 185-195: Not all renewable water input to a basin that is in excess of the base environmental flow requirement is available for abstraction (withdrawal), as a portion comes during floods that exceed the capacity of water impoundments. Table 1 states that "the outputs [of runoff and groundwater recharge] are temporally processed for further use" but it isn't clear what this means. This topic comes back up in the paper in line 455 so it seems to have been considered. Is there any reduction in the "Fr" to account for this aspect of renewable water supply?**

Thank you for asking for the clarity on the modelling of water resources. To address this concern, we would like to clarify that environmental flows are included in our model's water balance equation.

The model variable "Fr" is indeed used to represent this consideration. The renewable water supply (Fr) introduced into each basin is modified by deducting the environmental flow requirements.

Regarding the statement in Table 1 that the outputs of runoff and groundwater recharge are "temporarily processed for further use," we recognize that its meaning may not be entirely clear. We had summarized these steps in the main text and also included a table for further clarification in the Supplementary section. For your convenience, we are including it below:

*On the supply side, we use global gridded runoff and groundwater recharge data from the Community Water Model (CWatM)* (Burek et al., 2019) *and GHM outputs from ISIMIP* (Frieler et al., 2017) *Three bias-corrected meteorological forcing data outputs from different climate models (GFDL-ESM2M, HadGEM2-ES, IPSL-CM5A-LR) are used to estimate surface runoff and groundwater recharge. We use multi-model ensemble mean runoff and groundwater recharge as an available renewable freshwater resource. We aggregate the gridded data ($0.5° X 0.5°$ spatial, daily timestep) onto the B210 basins and 5-year annual average. For spatial aggregation, the spatial sum is used to sum the grid hydrological outputs (runoff and groundwater recharge) to the B210 basins. The detailed process has been summarized in Supplementary Table S2.*

Moreover, because we aggregate this water data onto annual time steps and the average could underestimate or overestimate the hydrological values, we have included the reliability of these supply data and the following text explains this approach:

*Regarding temporal aggregation, we apply a quantile approach with monthly freshwater (surface and groundwater) resources to incorporate hydro-climate variability and prolonged dry periods. For example, for the 10th percentile, the monthly mean is first calculated from daily data. Then we use the 10th percentile (Q90) of monthly freshwater sources, suggesting possible shifts in the energy and land sectors. We also test the scenario runs on the 30th (Q70) and 50th (Q50) percentile to see how these affects the model outputs.*

We hope this clarification addresses the reviewer's concerns and provides a more comprehensive explanation of how our model accounts for environmental flows and aggregates the gridded daily data from hydrological model outputs.

**Table S1** Steps used to process the gridded data within the nexus module

| Step No. | Scale | Input | Output | Procedure |
|---|---|---|---|---|
| 1 | Spatial | gridded | gridded | Convert kg/m2/sec to km3/ yr. and moving monthly average |
| | Temporal | monthly | monthly | |
| 2 | Spatial | gridded | basin | Spatial sum of grid values over basin |
| | Temporal | monthly | monthly | |
| 3 | Spatial | basin | basin | $Val_{2020} = (avg.\ val_{2015\text{-}2030\ rcp\ 2.6} + avg.\ val_{2015\text{-}2030rcp60.})/2$ |
| | Temporal | monthly | monthly (same 2020 value for rcp scenarios) | $Val_{2020}$ is applied to all data frames at this point. $Val_{2020}$ is monthly 2020 data |
| 4 | Spatial | basin | basin | Monthly bias correction is applied for each rcp value to adjust for the previous step. The bias correction is only done at 5 year intervals monthly data. For the 5 year average, MESSAGEix time step formulation is used such that for example ; $val_{2025} = (val_{2021} + \ldots + val_{2025})/5$ Now for $val_{2025}$, bias correction is done as; |
| | Temporal | monthly | monthly bias corrected | $delta_{rcp6} = avg.\ val_{2015\text{-}2030\ rcp\ 6} - val_{2020}$ $delta_{rcp2.6} = avg.\ val_{2015\text{-}2030\ rcp\ 2.6} - val_{2020}$ $val_{rcp\ 6\ bias\ corrected} = val_{rcp\ 6} + delta_{rcp6}$ $val_{rcp\ 2.6\ bias\ corrected} = val_{rcp\ 2.6} + delta_{rcp2.6}$ the delta is reduced by 0.2 in each 5 year interval until the delta reaches zero in 2045 |
| 5 | Spatial | basin | basin | 5 year monthly data is prepared from monthly bias corrected data by just filtering the 5 year timesteps (2020,2025,…2100) from the previous step |
| | Temporal | monthly bias corrected | 5 years monthly | |
| 6 | Spatial | basin | basin | Three reliability scenarios are created. $val_{q50}$, $val_{q70}$, $val_{q90}$ by taking quantiles of monthly bias corrected data. |
| | Temporal | monthly bias corrected | 5 years annual | |

**RC2.9.** **Line 230: Why is GDP a predictor for desalination capacity? I know a reference is provided but the relationship is not intuitive and should be explained here.**
**Line 238: Why is desalination potential influenced by climate? It seems that the infrastructural investment should be a function of climate, but this doesn't influence the desalination potential of any basin, which should be unlimited for coastal basins, and perhaps based on saline aquifer volumes in endorheic basins. This might just be a case of terminology, and "desalination potential" needs to be defined (normally a resource "potential" means the upper limit of production of the resource).**

Regarding the connection between GDP and desalination capacity, we recognize that it may not be readily apparent. In our study, the GDP served as one of the estimators for desalination capacity. While the precise nature of this relationship can vary, in general, higher GDP levels are associated with greater investment capabilities. These factors can aid in the growth and development of desalination infrastructure, while the governance indicators of the regions may not be able to fully utilize this desalination infrastructure. Considering these socio-economic parameters, water stress also plays a role

here, which means regions with higher water stress require more desalination.

Regarding the influence of climate on desalination potential, we understand the terminology's potential for causing confusion. In our research, "desalination potential" refers to the maximum desalination production capacity within each basin. This capacity is affected by several factors, such as the availability of saline aquifers in endorheic basins and the suitability of seawater as a feedstock in coastal basins.

**RC2.10.    Section 3.3 (SDG section): there's no mention of the internal conflicts within and between the SDG's analyzed, and how those are resolved in the scenario design. One pertinent example for this study is that SDG6 simultaneously calls for reducing water impoundments to restore natural aquatic environments (6.6), whereas many of the other SDGs would require improvements in flood control to provide irrigation water for high and stable crop yields (SDG2), and to protect farms and infrastructures (urban, transportation, industrial; SDGs 8-11) from flood-related damages. How this study balances such competing goals should be described in the scenario design.**

`Regarding the internal conflicts within and between the analyzed SDGs and how they are resolved in the scenario design, we appreciate your insightful comments. In fact, it is crucial to recognize the delicate balance required between competing objectives, such as those pertaining to water impoundments and flood control. Due to the spatial and model-scale limitations of our research, however, it becomes difficult to incorporate a comprehensive representation of flood-related damages. Modeling such intricate dynamics is currently beyond the scope of our study due to the associated computational costs. Despite this, our research strategy seeks to examine the role of individual SDGs within the integrated system. We can evaluate the contributions of individual SDGs to the overall system dynamics by designing specific scenarios. This strategy is consistent with our updated model runs (explained in the result sections), which shed light on how sector-specific climate impacts affect the broader outcomes of the integrated system. In future studies, we plan to expand our research framework to include more climate extremes and related damages, including the complexities of flood control.

**RC2.11.    Lines 355-360: The authors make reference to conducting a regional downscaling case study of Zambia in the methods, but then the results don't have anything about Zambia, and the spatial maps don't have it disaggregated. Perhaps this part should be dropped from the methods altogether? I'm not sure what was intended to describe this in the methods but not the results.**

Thank you for drawing our attention to this. We appreciate your suggestion to eliminate this section entirely from the methods to avoid misunderstandings. The purpose of providing this example was to demonstrate the adaptability of the model's application to other use cases, particularly at the national level. The mentioned example is distinct from the global scale model, but it describes how this methodology was used to generate a national prototype for Zambia.

**RC2.12.    Lines 436-454: I found these 20 lines in particular to involve lots of re-stating of obvious stuff that anyways isn't established by this study. The point being made is pretty simple: climate change impacts on precipitation patterns and therefore renewable water availability are heterogeneous both geographically and temporally, and this drives the behaviors in the integrated scenarios.**

We understand the ambiguity that was caused by the previous text in the Results section. While updating the section we have revised the following paragraphs and added quantifications of results across scenarios.

*The results demonstrate that renewable surface water and groundwater are limited and vacillate across different climate scenarios. These effects result in a decrease in renewable water consumption, which is more evident in the land sector than in the water sector. In addition, our model indicates an increase in the use of alternative water sources such as brackish water, effluent, and desalination in certain regions, indicating that renewable water resources are limited in these areas. These observations serve to further highlight the significance of the SDGs. For instance, when aligned with SDG 6 targets, the model predicts a 24% reduction in water consumption, resulting in a more sustainable allocation of water to environmental flows.*

*The geophysical characteristics and land use effects of various locations have a significant impact on the global effects of climate change on the water sector. Some areas may obtain benefits, while others may suffer negative consequences. In addition, the study found that the adaptive response to*

*climate impacts reduces by an average of 11% the number of individuals exposed to hunger. Compared to the SDGs (30%), where specific assumptions were made to reduce the danger of hunger, this reduction is less significant.*

**RC2.13.     Line 455: please see comment about lines 185-195; it isn't clear to me in the methods whether such "unavailable" water is deducted from the water supply in the model. It's kind of a tricky thing to model because the unavailable fraction is a function of infrastructure which in this modeling scheme is endogenous.**

Thank you for highlighting this confusion. We have removed this part to avoid ambiguity and presented results in a different manner.

**RC2.14.     Line 467: "there have been numerous publications on integrating SDG dimensions into Integrated Assessment Models" - please provide appropriate references.**

Thanks for highlighting this sentence where we were missing references. We have added appropriate citations in the updated manuscript.

*While there have been numerous publications on integrating SDG dimensions into Integrated Assessment Models (IAMs) such as (van Soest et al., 2019; Baumstark et al., 2021; Vuuren et al., 2022; Soergel et al., 2021a; Binsted et al., 2021), this study stands out due to its novel approach of combining SDG policies with climate goals and impacts and evaluating their effectiveness in understanding the climate adaptation needs.*

**RC2.15.     Line 467: "this study stands out due to its novel approach of combining SDG policies with climate goals and impacts and evaluating their effectiveness in understanding the climate adaptation narrative" - This research question has been discussed in Moallemi et al. (2022, One Earth) and in Soergel et al. (2021, Nature Climate Change). The approach used in this study still adds to the portfolio of methods and research questions that have been published about the nexus between the SDGs and climate change mitigation in process-based integrated assessment models, but the authors should make reference to the recent literature and clarify what the present study adds. Also worth reading is van Vuuren et al. (2022, One Earth) and van Soest et al. (2019, Global Transitions)**
**I agree! The Results & Discussion section would be a great place to conduct such a comparison. All of the key results shown in Figures 5 and 6 should be compared against existing literature. A lot of the value added of this study is the integration of multiple different systems, but most of the variables have also been assessed in other studies.**
**Figure 6 - These are strange and counter-intuitive results (for me, anyway) that should be described, and also should be indexed against the total electric supply in each place. On its own, 100 TWh of electricity in some region in some year far into the future is a pretty abstract thing; even if a reader happens to know the base-year electricity supply in these different macro-regions, the values far into the future for some scenarios are not necessarily known. Within the data shown, one interesting thing is the net change in total electricity demand. There are particularly large increases in "other non-fossil" generation which in some cases more than counterbalance the reduction in fossil generation that is due to climate impacts. Why is this?**
**If the climate-driven increase in total electricity supply is demand-driven, is this all because of additional air conditioning demand? If so, how do the results here compare with the existing literature, e.g., Clarke et al. (2018, Energy Economics) or van Ruijven et al. (2019, Nature Communications)? These quantities of net electric demand increase (driven by climate impacts, all else equal) seem large at a glance, but again without the sectoral decomposition and presentation of baseline electric demand for cooling, it's hard to interpret, and anyways I can't recall exactly what the literature estimates for climate-driven growth in electricity demands for air conditioning in buildings.**
**Also, there should be observational studies on the climatic sensitivity of fossil generation to temperature changes, and just at a glance these results seem to be more climatically sensitive than what I'd expect. Regardless, my expectations shouldn't matter; the results should be compared with the literature. I believe Michelle van Vliet has published a few papers on the topic that could be useful as a starting point.**

**Another question that figure 6 brings up is whether these changes reflect changes in investment over time, versus capacity factors (i.e., operational differences for a similar capital base). The increase of ~80 TWh in North America hydro stands out here; is this from pushing capacity factors up due to increased water flow through the existing hydropower installed capacity, or is this from new hydropower investment that is encouraged by increased river flows and/or increased market prices for electricity?**

Thank you for your valuable comments and suggestions. We have taken them into consideration and made significant improvements to the results and discussion section. Specifically, we have restructured the section to provide meaningful indicators and comparisons with the existing literature. We have included a table that presents global values of key water indicators and compares them with relevant studies. Furthermore, we have updated Figure 5 to demonstrate the water balance components and flows, highlighting the model's capability to report results at different stages of the water balance. Figure 6 now depicts the variation of key indicators over the time horizon, including the changes when climate impacts are incorporated into the model. We have also added a Supplementary Figure (S4.1.4) showcasing a Sankey diagram of water sector flows throughout the model. These additions, along with the accompanying explanations in the manuscript, aim to address the counter-intuitive results and provide comparisons with the literature. We appreciate your suggestion to consider the climatic sensitivity of fossil generation and the investment versus capacity factor dynamics, and we will explore the relevant literature, including the work of Michelle van Vliet, to provide further insights. Following is the updated text in the revised manuscript:

*The scenario formulation we used to describe results are.*

[revised manuscript text omitted]

---

## Author Response (AR2)

**Reviewer 1**

**Thank you very much for the substantial revision and the clarifying responses to my questions. The paper has improved a lot, reads much better and describes many aspects much more clearly. However, there is still some need to improve the results section from my point of view as well as a list of mostly language-related points.**

We appreciate your recognition of the improvements made in our previous submission. We made additional improvements to the results section to improve clarity and coherence. Specifically, we separated the Results and Discussion section to clarify the interpretation and results. We have also refined the linguistic components, encompassing grammar and syntax. We value your valuable feedback and believe that these updates effectively address your issues, hence improving the overall quality of the paper. The original comments are in **bold**, our comments are in standard text, and the additions/revisions are indicated in *Italics*.

**Specific comments**

**RC1.1.      Most importantly, the results section is still not very strong in terms of content. This maybe be due to the preparation of a number of other papers which will go much deeper into applications of the NEXUS module. That is fine, but the current state is still something which feels quite incomplete and repeats in some parts model description rather than discussing results (e.g. from line 600 to line 627). Figure 5 is very nice but not discussed in detail. The discussion mentions an impact on the power generation mix for which a figure would be nice (or is that Figure S4.2.1?). If no detailed scenario analysis is supposed to be performed in this paper, maybe the discussion can be framed around demonstrating that the integrated framework works, with selected plots showing the interactions. I feel this is what you are trying to do already, but from the current text this doesn't become clear.**

We appreciate your insightful comments on our manuscript's Results section. We appreciate your recognition of Figure 5 and acknowledge your concern regarding the level of detail in the current presentation of the results.We have changed the Results and Discussion sections to improve their completeness and clarity in response to your feedback. To give a clearer and more focused narrative that highlights the results of our investigation and the operation of the Nexus module, we have divided these sections.To avoid repeating the model description, we have also added further explanations of the results and enhanced the discussion with more in-depth interpretations of Figure 5. We have also examined the influence on the mix of electricity generation, ensuring that the relevant statistics are adequately explained.

The specific addition of the paragraphs mentioning the results are:

*Figure 5 presents a comparative analysis of key Energy-Water-Land (EWL) indicators across a spectrum of modeled scenarios. The boxplot distributions visually depict selected model output indicators for the period from 2030 to 2080, covering scenarios such as Reference, Impacts, Impacts_LU (land use), Impacts_EN (energy), Impacts_WAT (water), and SDGs. The graph's*

*constant trend in energy-related metrics across scenarios stands in stark contrast to the pronounced unpredictability of non-renewable water usage, suggesting that energy indicators are less vulnerable compared to water and land.*

*Figure 5 also shows that, despite the biophysical impacts, agricultural production doesn't vary much. The SDG scenario, however, results in a considerable 20% decrease in agricultural output, with the biophysical implications of land usage having a particular influence on sugar crop yields. This noteworthy effect emphasizes how susceptible some crops are to changes in land use and how crucial it is to take these effects into account when developing agricultural plans and policies.*

*Furthermore, the primary cause of the decrease in water withdrawals is the consequences of land use, wherein $CO_2$ fertilization effects are a major factor. These effects on land usage decrease the overall need for irrigation and increase the efficiency with which agricultural operations use water.*

*Additionally, the figure also indicates that the cost of potable water has increased by 80%, primarily due to the adoption of environmental flow allocations aimed at protecting freshwater ecosystems and the increased expenses linked to sophisticated wastewater treatment procedures. These elements highlight the intricate relationship that exists between water resource management and economic results as well as environmental care.The geophysical features and land use influences of various regions mostly determine the global consequences of climate change on the water sector, with certain areas experiencing gains while others may have negative effects. Adaptive responses to climatic impacts reduce the number of people exposed to hunger by an average of 11% according to the study. This is not as significant as the 30% reduction in the SDG scenario, which is based on specific actions to reduce the risk of hunger.*

*It is imperative to exercise caution when interpreting the outcomes of the different scenarios, taking into account their reliance on several assumptions and their suitability for particular geographical and temporal circumstances. However, these results offer insightful information about the possible financial effects of various water management techniques. Different modeling methodologies may produce different results because assumptions, data inputs, and other elements are inherently variable. It is feasible to determine the most effective and successful tactics and to obtain a more thorough understanding of the probable consequences of different water management systems by comparing the outcomes from many models.*

**RC1.2.      The scenario selection is still not very clear, in particular whether all impact scenarios use RCP6.0 and if so where RCP2.6 comes in. Maybe only in the SDG scenario? But in Figure 5 fossil energy production is very similar across scenarios. Does the SDG**

**scenario include impacts? Also, I assume the figures in the supplement will be updated to the new scenarios?**

We have clarified the scenario framework in the manuscript and added a table for clear categorization. The SDG scenarios include climate impacts, with the intent to assess the combined effects of climate impacts and SDG implementation.

The revised section in the manuscript is;

*In our analysis, we have currently applied the SSP2 framework in conjunction with both RCP2.6 and RCP6.0 to establish the current model setup. Future work will incorporate a broader array of SSPs paired with various RCPs to ensure a more comprehensive and coherent set of assumptions across different scenarios. Our examination of the biophysical effects of climate change on energy, water, and land use sectors involved contrasting scenarios that integrate climate impacts—specifically designated as Impacts, Impacts-EN (focusing on the energy sector), Impacts-WAT (water sector), and Impacts-LU (land use)—alongside SDGs. We measured these against a Reference scenario, which is predicated on historical climatic patterns and excludes any projections of climate impacts or SDG considerations. The scenario assumptions are detailed in Table 4.*

***Table 4*** *Summary of Scenario assumption*

| Scenario | Climate Scenario | SDGs |
|---|---|---|
| *Reference* | *Historical climate assumptions for RCP 6.0 across EWL sectors.* | *Not included* |
| *Reference (Mitigation)* | *Historical climate assumptions for RCP 6.0 across EWL sectors.*

*This scenario, although is practically not feasible it is used to compare the responses of the new features* | |
| *Mitigation* | *RCP 2.6 (biophysical impacts of EWL sectors as outlined in Table 2 and section 3.2 )* | |
| *Impacts* | *RCP 6.0 (biophysical impacts of EWL sectors as outlined in Table 2 and section 3.2 )* | |

| | | |
|---|---|---|
| Impacts_LU | RCP 6.0 (biophysical impacts of land sector, e.g. crop yields) | |
| Impacts_WAT | RCP 6.0 (biophysical impacts of hydrology) | |
| Impacts_EN | RCP 6.0 (biophysical impacts of energy, e.g., cooling demand and renewable potential) | |
| SDGs | RCP 6.0 (biophysical impacts of EWL sectors as outlined in Table 2 and section 3.2 ) | SDG 2, 6, 7, 13, 15 – as outlined in Table 3 and section 3.3 |

**RC1.3.    Finally, the introduction could still be improved to state more clearly the gaps closed by this study. It should be pointed out explicitly that adaptation here means model-endogenous responses to impacts or SDG constraints, no adaptation pathways with specific adaptation policies.**

We have refined the introduction to clearly delineate the study's contributions in addressing existing research gaps. Additionally, we have specified within the text that 'adaptation' refers to model-endogenous responses to climate impacts and SDG constraints, without prescribing specific adaptation policies at the community level. We have also updated references and introduction to clarify more. The changes are reflected in the tracked version of submission at various sections.

**Minor points (the line numbers are from the "version 3", not the track changes version of the paper)**
Thank you for highlighting the minor comments. We have mentioned the revisions under each minro comment where needed.

**Line 64: "IAMs often consider the costs of resources in an aggregate spatial region" – not only costs of resources but this is their level of spatial resolution**

**Line 65: "the key element for change is implemented on a local/national scale"**

The above two comments are addressed by rewriting both sentences as:

*IAMs typically operate at regional or continental scale for informing the future pathways, whereas adaptation strategies requires a more nuanced, localized focus emphasizing on national and sub-national levels (Andrijevic et al., 2023).*

**Line 71: there seems to be a remnant sentence left over, starting with "estimating that economic impacts have…"**

The sentence has been revised as;

*These sectoral assessments evaluate biophysical impacts such as changing yields, runoff changes, food production, and groundwater. Economic impacts are subsequently estimated using various methodologies, chosen based on the specific type of impact considered such as the corelation between climate damages and temperature variations.*

**Line 77: The sentence starting with "It is becoming quite evident" should be revised in terms of language as it does not read well. Also, the statement on "the effects of different sectors on the techno-economic outlook" is unclear – do you mean climate impact channels and techno-economic transformation or scenarios or so?**

We have revised these setences as;

*Some studies have empirically linked climate conditions with socioeconomic systems and incorporated distributional factors into cost-benefit models, resulting in increased social costs of carbon and more stringent mitigation pathways (Parry and Carter, 2019; Howard and Sterner, 2017). Incorporating the representation of biophysical climate impacts into integrated assessment models is crucial to understand how various sectors influence techno-economic scenarios and to identify appropriate mitigation and adaptation strategies (van Maanen et al., 2023; Andrijevic et al., 2023)*

**Lines 80-84: Piontek et al. 2021 a and b are the same paper. Maybe you mean Schultes et al. 2021, which does include aggregate damage functions but not biophysical impacts. Soergel et al. 2021 does not include climate impacts at all. There are probably other papers to be mentioned here, certainly from the CGE literature which capture impacts in the water, land and energy sectors?**

We have improved this section as;

*Incorporating the representation of biophysical climate impacts into integrated assessment models is crucial to understand how various sectors influence techno-economic scenarios and to identify appropriate mitigation and adaptation strategies (van Maanen et al., 2023; Andrijevic et al., 2023) . (Piontek et al., 2021) analysed the economic impacts of climate change using the REMIND IAM model, but biophysical climate impacts were not represented. (Soergel et al., 2021a) emphasized the significance of considering the consequences of climate impacts and evaluating how integrated scenarios respond to these impacts, especially regarding sustainable development pathways.(Schultes et al., 2021) highlights the economic impact of climate change, advocating for immediate mitigation to reduce long-term damages and align with cost-effective Paris Agreement targets*

**Line 83: What do you mean by "introducing climate impacts in the development trajectories"? As far as I know Taconet et al. do a damage post-processing using damage functions?**

We have removed this sentence to keep the flow and consistency of the Introduction.

**Lines 89: Yes, so you can state here clearly this need requires biophysical impacts to be included in the IAMs, preferably with a link to high resolution impact modeling – and that is what you deliver!**

We have improved the sentence as;

*This study proposes a framework that incorporates high-resolution model outputs of biophysical climate impacts into IAMs, strengthens the water sector's resilience, and crafts scenarios in tandem with sustainable development objectives to evaluate climate change effects across various pathways, including mitigation, adaptation, and sustainability.*

**Line 98: You mention communities but IAMs don't resolve communities.**

We have improved the sentence as;

*This holistic approach is designed to elicit model endogenous response to climate impacts and SDGs constraints, thereby enhancing systemic resilience and advancing sustainable development, although it does not delineate specific adaptation policies at the community level.*

**Line 99: "This study addresses these gaps" – maybe start the paragraph with something like "This study addresses the following gaps"? It is not immediately clear from the flow.**

We have improved this section:

*Due to hydrological data's spatial and temporal complexity, it is challenging to translate hydrological information into the IAMs. Usually, the spatial extent of IAMs is macro-regions, and the aggregated hydrological information loses adequate information at a macro-level. There is 105 always a need to find a middle ground between showing the hydrological process more accurately and lowering the cost of computing  (Fricko et al., 2016b; Parkinson et al., 2019b)There have been efforts to link a higher spatial resolution water sector to account for hydrological balance and constraints in IAMs, such as (Yates, 1997) and (Kim et al., 2016).*

*Addressing the identified gaps, this study proposes a framework that integrates climate impacts with an emphasis on the water sector's role in climate change and develops scenarios in sync with sustainable development assumptions to evaluate the effects of climate change within the contexts of mitigation, adaptation, and sustainable development pathways.*

**Line 143: "the core global framework" – of MESSAGEix?**

We have adjusted this part.

**Line 168: "This enables connecting …"**

We have fixed this sentence.

**Lines 193/194: "policy options", not "possibilities" – also "many facts and hypotheses" is somewhat vague, many of the examples mentioned are scenario-based**

The framework facilitates a comprehensive assessment of policy options by integrating scenario-based projections, including population and economic growth, technological advancements, and resource limitations.

**Lines 205-207: post-processing, not post-computing? With "scenario explorer" do you mean the IIASA scenario explorer? Maybe not all readers are familiar.**

iv) post-processing of the model outputs to provide ready-to-use results in a database and for visualization tools such as IIASA scenario explorer (Huppmann et al., 2018).

**Line 211: R11 region – please define**

The module uses SSP-RCP (Shared Socioeconomic Pathways – Representative Concentration Pathway) combinations as narratives for creating a baseline scenario. Each scenario is developed using SSP-RCP combinations, national policies, and Sustainable Development Goal (SDG) assumptions aggregated at the R11 region, a spatial delination of 11 global regions used in the MESSAGEix-GLOBIOM.

**Line 215: you use GHM output from the ISIMIP database, correctly? In the current wording, it seems to be two separate things.**

We used the Global Hydrological Models (GHMs) outputs from ISIMIP database (Frieler et al., 2017) for water availability and hydropower potentials for biophysical impact indicators. The GLOBIOM model upscales these water requirements and provides irrigation requirements at an aggregated 37 regions based on land-use allocation decisions

**Line 242: Do you mean Figure 1? Also "The study represents" – maybe "The study applies"?**

We have adjusted the wording here.
**Line 292: delete "driving"**

We have deleted the word 'driving'.

**Line 312: maybe state explicitly that the data from Wang & Sun are based on the SSPs for the readers unaware**

We followed the methodology by (Graham et al., 2020) to estimate the municipal water demands, where urban and rural components are derived from gridded population and income-level projections based on the SSPs, as detailed in(Wang and Sun, 2022).

**Line 323: Figure 1?**

We apologize for the confusion. Figure 3 was missing in the previous version by mistake. We have re-added in the updated version.

**Line 395: What about bioenergy as renewable energy source? That is affected through impacts in GLOBIOM, correct?**

We don't take into account the impacts on the bioenergy in the current framework.

**Line 411: "no climate change scenario" – REFERENCE?**

We have removed these words to avoid confusion since scenarios are defines afterwards in the manuscript.

**Line 463: "protect river-related ecosystems in alignment with achieving SDG target 6.6"?**

We have adjusted the sentence as;

*Maintaining environmental flows in rivers is instrumental in achieving SDG target 6.6, which aims to protect and restore water-related ecosystems, encompassing a range of natural landscapes from mountains and forests to wetlands, rivers, aquifers, and lakes.*

**Lines 565/566: Maybe clearer to say that the reference scenario keeps climate constant at historical levels? The phrasing "physical impractical" is strange, the reference scenario is unrealistic given progressing climate change as well as already implemented mitigation and adaptation measures.**

We have added a note in the scenarios Table 4 that scenario is mere included for model validation and assessment against the biophysical impacts response.

**Lines 571-575: This should come in the model description, not in the discussion of results.**

We have moved this part to Discussion section.

**Lines 576: Maybe introduce this section better, i.e. by stating that you do a bit of model validation by comparing 2020 values with other sources to help guide the reader.**

We have improved the introductory section as;

*Our study presents detailed results of water balance flows, providing a critical examination of global water management and the interdependencies within the water, energy, and land nexus. By comparing our model's outputs with benchmark values from the literature, we establish a validation baseline for EWL indicators, ensuring our findings resonate with recognized global*

*estimates. Our study allows the monitoring of water balance flows at varying stages, offering an in-depth understanding of global water management and the intricate nexus between water, energy, and land. These interactions are depicted in Figure 5a in form of Sankey diagram, along with input details and assumptions expounded in Section 3.1.*

**Line 635: do you mean in the SDG scenario?**

We have adjusted this as SDG scenario.

**Line 638: which RCP is underlying this result?**

RCP 6.0 is being discussed here.

**Line 652: What adaptive responses do you mean here exactly? This relates to the need mentioned above to state somewhere in the beginning more clearly how you address adaptation in this study.**

We have adjusted the sentence as;

*In addition, these effects contribute to a 28% decrease in the marginal price of potable water due to adaptive responses to climate change impacts in electricity and irrigation withdrawals.*

**Line 660: What do you mean by climate scenarios? Rather transformation or mitigation scenarios?**

We have adjusted the wording as;

*Overall, the findings indicate the need for more research to fully comprehend the potential effects of climate change on diverse sectors and the possibility that incorporating biophysical consequences can substantially impact the outcomes of Impact and Mitigation transformation scenarios.*

**Line 663: Can you provide an example for co-benefits?**

We have added example of co-benefits in the sentence;

*Overall, the study's findings illustrate the significant implications of climate impacts in mitigation scenarios on the energy mix and the co-benefits such as agriculture production, increased crop yields, shift towards less fossil intensive technologies in electricity mix*

**Lines 685/686: This sentence is repeating something stated earlier.**

We have removed this part.

**Line 722: Isn't consistency in the impacts why you chose ISIMIP? It is a bit unclear what you mean here.**

We have improved the sentence to provide more clarity;

*While the Nexus module employs the robust outputs of the ISIMIP for depicting climate impacts, there are certain challenges from the current set of outputs not being fully consistent with the input climate scenario assumptions. As soon as updated and aligned ISIMIP outputs become available, we will conduct a new model run to enhance consistency and reduce uncertainty in our analysis.*

**Line 729: What do you mean by this sentence? Which complex dynamic isn't captured?**

We have explained which dynamics are not captures;

*The current model structure, which assumes an endogenous adaptation response, may not fully capture the complex dynamics such as the feedback mechanisms between water availability and energy production, socioeconomic impacts of water scarcity on land use, and long-term societal adaptations to water stress within the EWL sectors. Future research will focus on integrating these inter-sectoral feedback and dynamic responses to enhance the model's accuracy in depicting the intricacies of the EWL nexus.*

**Line 732: What do you mean by "higher tolerance"?**

We have adjusted the wording to explain better;

*In future research, we plan to expand our exploration of climate impact dimensions to include a more robust handling of statistical climate extremes, aiming for greater resilience in our model's performance at sub-annual temporal resolutions. Future versions of the model will integrate up-to-date climate impact data and strive for more consistent data sources across sectors.*

**Reviewer 2 (Page Kyle):**

**The authors have addressed my concerns from the prior review, and I appreciate the many comparisons to the literature for the selected key results. However in re-reading the final document I noticed a large number of minor issues that should be corrected prior to finalizing the document for publication. These are presented using the line numbers in the clean document egusphere-2023-258-manuscript-version3.pdf.**

We thank for acknowledging the improvements made to the manuscript in the previous round. We have addressed the minor issues you have highlighted and ensure the document is thoroughly refined.

**RC 2- Minor Comments**

**42 - copy editing error (two periods)**
**77, 80, 82, 88 - each one of these lines has a copy editing error related to references, parentheses, and punctuation.**

The copy-editing errors on lines 42, 77, 80, 82, and 88 have been thoroughly addressed and corrected in the revised manuscript.

**88-89 - SSP acronym used in line 88, defined in 89**

We have adjusted this.

**90-105 - note that the "tracked" and "clean" version differ here**

Sorry for confusion. Apparently, there was some technical issue overlooked between two versions in the last round.

**94-95 - the EWL acronym is used undefined on line 94 and then written in long form and with capital letters (which I don't think is correct) on line 95 but without providing the acronym. I'd write: "meeting population-driven demands in the energy-water-land (EWL) sectors (Rasul and Sharma 2016). Integrating cross-sectoral EWL nexus..."**

We have adjusted the acronym and sentence as per your suggestion;

*Integrating cross-sectoral EWL nexus analysis in IAMs can help identify trade-offs and synergies, integrate policy implementations, and address equity dimensions, such as the population exposed to hunger or lacking access to sanitation and electricity*

**261 - please replace "South Africa" (which is a country) with either "Sub-Saharan Africa" (which is what SSA stands for in MESSAGE) or "Southern Africa".**
We agree with your observation and have replaced the South Africa with Sub-Saharan Africa where applicable.
**307 - I'm not sure what is meant by "Figure 2.1 B" as there is no such figure in the supplement.**

There was an 'S' missing and we have adjusted in the manuscript.

**577 - "approximately 47219.79 km3/yr" - if the text says approximately, then the number should not have 7 significant digits.**

We have adjusted all figured to neares 1 km3.

**610-611 - the example information flow described is introduced as "The use of EPIC" but then the information flow involves more than EPIC. Perhaps "The use of EPIC and GLOBIOM in determining irrigation responses and crop yields is one prominent example."**

We have rewritten this section to have more clarity;

*To capture the dynamic responses of the climate system, our model employs a multifaceted strategy that leverages both endogenized and exogenous outputs. Specifically, we utilize EPIC to obtain insights on irrigation responses and their subsequent effects on crop yields, which is a key example. These yield outputs from EPIC are then integrated into GLOBIOM, where adaptation responses are internalized, prompting a strategic reallocation of land use resources in response to climate impacts.*

**632 - does "fertilization intensity" here refer to chemical fertilizers (N, P) or CO2 fertilization? Either way it's not spelled out (and it should be) how the fertilization intensity influences water use efficiency and therefore irrigation water requirements.**

We have clarifies the fertilization intensity;

*The effects of climate on crop yields show variability, with sugar crops experiencing a significant impact at 16%, while cereals exhibit a comparatively modest change of approximately 1%. The net yield effect is directly influenced by the intensity of nitrogen and phosphorus fertilization, which enhances water use efficiency and consequently reduces the demand for irrigation water. Furthermore, in our climate impact scenarios, increased CO2 levels also increase crop yields and contribute to improved water use efficiency, which is factored into our results.*

**665 - "these results are based on a specific model and situation and should be interpreted as a general trend" - the point being made seems to be the opposite, that it's only one model so should not be interpreted as a general trend. But I'm not sure of the point being made so the authors should check.**

We removed this sentence to avoid confusion .

**681 - I've found that throughout the paper, the terms "module" and "model" used inter-changeably and it's often confusing and hard to tell what is intended. I would think that "model" would refer to the integrated system of individual "modules", where module means component (e.g., GLOBIOM, MESSAGEix, MAGICC, and so on).**

**763-764 - here again the terms "model" and "module" seem to be used interchangeably but I don't know if that is intended.**

We understand the choice of terms ' module' and model' could be confusing to readers. We have adjusted all the work related as 'modules'. The changes are reflected in the revised manuscript.

**753-760 - can the authors confirm that these results are described or depicted elsewhere in the study? I don't see it, and the conclusions shouldn't be the first place where a result is described.**

We have removed this part to avoid confusion and make the text consistent.

---

## Author Response (AR3)

**Reviewer**

**Thank you for addressing all previous comments so thoroughly! The paper reads well now and I recommend it for publication. However, I suggest one last round of careful proof-reading. In particular, in some of the text which was moved around to another part of the paper, line numbers were left in the text.**

We appreciate your recognition of the improvements made in our previous submission. We made additional  minor revisions to the manuscript after proofreading.